

# The rise of obfuscated Android malware and impacts on detection methods

Wael F. Elsersy, Ali Feizollah and Nor Badrul Anuar

Department of Computer System and Technology/Faculty of Computer Science
and Information Technology, Universiti Malaya, Kuala Lumpur,
Wilayah Persekutuan Kuala Lumpur, Malaysia

## ABSTRACT

The various application markets are facing an exponential growth of Android malware. Every day, thousands of new Android malware applications emerge. Android malware hackers adopt reverse engineering and repackage benign applications with their malicious code. Therefore, Android applications developers tend to use state-of-the-art obfuscation techniques to mitigate the risk of application plagiarism. The malware authors adopt the obfuscation and transformation techniques to defeat the anti-malware detections, which this paper refers to as evasions. Malware authors use obfuscation techniques to generate new malware variants from the same malicious code. The concern of encountering difficulties in malware reverse engineering motivates researchers to secure the source code of benign Android applications using evasion techniques. This study reviews the state-of-the-art evasion tools and techniques. The study criticizes the existing research gap of detection in the latest Android malware detection frameworks and challenges the classification performance against various evasion techniques. The study concludes the research gaps in evaluating the current Android malware detection framework robustness against state-of-the-art evasion techniques. The study concludes the recent Android malware detection-related issues and lessons learned which require researchers' attention in the future.

## INTRODUCTION

Since the advent of Android systems, smartphone devices are seen everywhere with a market share of 87% (*Chau & Reith, 2019*). Hence, Android devices have become the most popular devices for hackers and malware authors to target. With many open-source libraries in Android, Android application development tools enable young developers to develop Android malware applications. Therefore, the number of Android malware increases exponentially. In the Google Android market, Android applications exponentially grow from 2.8 million as of September 2018 (*Statista, 2016*, *2021*), to almost double, to reach 3.4 million apps as of the first quarter of 2021 (*Statista, 2021*). Nevertheless, Android malware authors attract end-users using cracked games, free applications, and video downloader applications. They mainly aim to spy on private data (*e.g.*, contact lists, photos, videos, documents, and account details) or control devices by

Corresponding authors
Wael F. Elsersy,
wfarouk@siswa.um.edu.my
Nor Badrul Anuar,
badrul@um.edu.my

remote servers as botnets (*Karim et al., 2015*). Android applications use Java as a developing language because Java provides a very flexible code, dynamic code loading (*Liang & Bracha, 1998*), and many other features to make Android application development more accessible and efficient. Likewise, Java uses obfuscation tools (*Aonzo et al., 2020*; *GuardSquare, 2014*) to protect commercial software companies from software plagiarism issues; professional developers protect their source codes from being stolen using advanced evasion techniques (*Aonzo et al., 2020*) as protection mechanisms. However, malware authors use the above-mentioned advanced Java features and evasion tools to reproduce more sophisticated Android malware, evading professional anti-malware (*Preda & Maggi, 2016*). Google introduced Google Bouncer (*Rahman et al., 2016*); however, Android malware successfully defeats Google Bouncer using different evasion techniques (*Maiorca et al., 2015*). Furthermore, Google Play Protect (*Xu et al., 2016*) service is the default device protection tool available on Google Android from Version 6.0 onwards; however, the previous versions are deprecated.

The rationale behind this study is the ability of evasion techniques to hinder the analysis process and thus the detection of Android malware. In 2021, PetaDoid (*Karbab & Debbabi, 2021*) proposed Android malware detection using deep learning techniques. PetaDroid builds static analysis Android malware detection framework using a 10 million Android apps dataset. PetaDriod addressed obfuscations in his study and concluded in his experimental results that his trained machine learning model that reaches 99.2% using static analysis would not detect complex obfuscated malware applications. The complex obfuscation techniques defeat Android malware detection PetaDroid model, which results into false detection. Though PetaDroid focused on trivial and some non-trivial obfuscation techniques. PetaDroid admitted that further deep analysis is required to address the sophisticated obfuscation techniques. The study focused on several evasion techniques, such as package transformation, string encryption, bytecode encryption, code obfuscation, injecting new codes *via* dynamic code loading, junk/dead code injection, emulation detection running sandboxing, and user interaction emulation detection. Android malware modifies the package, developer signature, and other information using the repacking evasion technique.

Moreover, the availability of various evasion techniques to the malware attackers increases the fear of developing very advanced obfuscation techniques, as such newly developed malware applications adopt advanced obfuscation techniques. It creates a challenge between preventing source code piracy and malicious attacks (*Gurulian et al., 2016*; *Zhang et al., 2014*) and struggling to decompile the malware application packages for further analysis (*Gonzalez et al., 2015*). Android malware detection frameworks (*Arp et al., 2015*; *Elish et al., 2015*; *Poeplau et al., 2014*; *Chen et al., 2015*) suffer from False Negative (FN) detection, which means the Android malware detection frameworks fail to detect some malware applications. The main reason behind FN is the malware evasion techniques that malware applications adopt to hinder detection. For instance, *Arp et al. (2015)* achieved 94% detection accuracy because it fails to detect malware with dynamic code loading transformation, one of the advanced evasion techniques. Likewise, *Elish et al. (2015)* used trigger-based dependence for privileged API calls, but it is unable to detect

malware families with code obfuscation and reflection transformation. *Poeplau et al. (2014)* used the call graph methodology to detect malicious code loading, and the native code dynamically loads the code.

Similarly, *Chen et al. (2015)* identifies a repackaged application in 10 s using code graph similarity but is incapable of tracking junk code insertion transformation. *You & Yim (2010)* reviewed the obfuscation technique, metamorphic and polymorphic malware types. They discussed the metamorphic and polymorphic evasion techniques; however, they neglected transformation and anti-emulation evasions. Furthermore, they merely reviewed evasion methods and failed to evaluate current evasion detection systems to evaluate whether they can detect evasive malware. *Sharma & Sahay (2014)* reviewed polymorphic and metamorphic malware and discussed their characteristics. They failed to mention evasion detection methods and evaluate the currently proposed methods. *Sufatrio et al. (2015a)* surveyed Android malware detection methods and briefly assessed a handful of related works in terms of evasion detection.

This study is intended for Android malware detection research highlighting the research gaps in malware detection caused by different evasion techniques. This study highlights the obfuscation and transformation techniques that need more attention from the research authors in future. It also provides guidelines and lesson learned to face this challenge. Due to the above facts, the authors take the challenge to introduce the following foremost contributions.

- We present evasion taxonomy, particularly in the Android platform. Our goal is to systematize the Android malware evasion techniques using a taxonomy methodology, which clearly shows various evasion techniques and how they affect malware analysis and detection accuracy.

- We scrutinise Android malware detection academic and commercial frameworks while a large portion of the past work concentrated on commercial Anti-malware solutions. This study examines different evasion techniques that hinder detecting malicious parts of applications and affect detection accuracy by reviewing state-of-the-art Android malware studies and issues limiting the detection of evasion techniques. It is worth noting that this work differs from related works that examine detection methods, as we go through evasion techniques that let malware eludes detection methods. Given the vast number in this study field, our investigation focuses on studies written between 2011 and early 2021 and innovative contributions that appeared in high-ranked journals or conferences such as IEEE, ACM, and Springer, hence the identified related papers are 511 research papers.

- We highlight the existing problems and gaps in Android malware evasion detection by examining the previous frameworks and identifying the Android malware evasion detection research gap.

- We introduce a decent number of recommendations and lessons learned to consider in future work around research. We also aim to highlight the contribution of each study, challenges, countermeasures, and open issues for future research.

**Table 1 Comparison of the recent reviews.**

| Related studies | Evasion techniques discussion | Evasion detection tools evaluation |
|---|---|---|
| This study | Encryption, package and code transformation, code obfuscation, anti-emulation | Commercial + Academic |
| Droidchameleon (*Rastogi, Chen & Jiang, 2013*) | Transformation | Commercial |
| Vikas (*Sihag, Vardhan & Singh, 2021a*) | Code Obfuscation, repackaging | Academic |
| FeCO (*Jusoh et al., 2021*) | Code Obfuscation, Encryption | Academic |
| Rastogi (*Rastogi, Chen & Jiang, 2014*) | Encryption + Transformation | Commercial |
| AAMO (*Preda & Maggi, 2016*) | None | Commercial |
| Hoffmann (*Hoffmann et al., 2016*) | Obfuscation | Commercial |
| Tam et al. (*Tam et al., 2017*) | Transformation + Obfuscation | None |
| Nguyen-Vu et al. (*Nguyen-Vu et al., 2017*) | Transformation | None |
| Kim et al. (*Kim et al., 2016*) | Anti-emulation | None |
| Xue et al. (*Xue et al., 2017*) | Encryption | Commercial |
| Bulazel (*Bulazel & Yener, 2017*) | Virtualization and performance case studies | Academic |

Table 1 presents the differences between this study and the recent evasions detection reviews. Vikas (*Sihag, Vardhan & Singh, 2021a*) evaluated the hardening code obfuscation tools against the reverse engineering process; however, it focused on development advantage more than malware detection perspectives. FeCO (*Jusoh et al., 2021*) focused on Android application static analysis and Android malware detection using machine learning and deep learning methods. It highlighted the type of code obfuscations techniques and previous research obfuscation solutions. AAMO (*Preda & Maggi, 2016*) and Droidchameleon (*Rastogi, Chen & Jiang, 2013*) study the effectiveness of evading commercial anti-malware applications by using their evaluation tools; Droidchameleon (*Rastogi, Chen & Jiang, 2013*) examines trivial transformation, which easily evades the detection of Android malware using the most popular anti-malware commercial packages. However, Droidchameleon (*Rastogi, Chen & Jiang, 2013*) misses studying the effect of the evasion techniques on current detection accuracy. Likewise, Rastogi continued his study of Droidchameleon (*Rastogi, Chen & Jiang, 2013*, *2014*) and added more composite transformation attacks that consist of more than evasion attacks and investigated evasion chains' capability for hindering malware detection. Hoffmann develops a tool to thwart malware detection and evaluates the accuracy of a few typical static and dynamic malware analysis frameworks and concludes that code obfuscation evasion evades Android malware detection frameworks (*Hoffmann et al., 2016*). Nevertheless, Hoffmann excludes some evasion techniques from the evaluation of malware detection frameworks.

The rest of the paper is organized as follows: the survey methodology and background section provide essential background information for this study; we explore the Android operating environment and its weaknesses. Evasion techniques section presents the evasion techniques taxonomy with regards to different categories of evasions. Android

evasion detection frameworks section investigates the current state-of-the-art evasion detection frameworks and evasion test benches tools. We discuss the lessons learned and future directions in discussion and lessons learned sections. Finally the last section represents the conclusion of this study.

# SURVEY METHODOLOGY

## Methodology

The methodology of retrieving Android malware obfuscation detection related articles is presented in this section. This study adopted Web-of-Science search engine to carry over the literature review using search terms with inclusions and exclusion criteria. The review process consists of four phases; first phase is identification, second phase is screening, third phase is eligibility, and fourth phase is analysis phase.

### Identification

The adopted Web-of-Science search engine covers hundred years of citation data containing many journals related to computer security, software development, and network security. Clarivate Analytics established this citation database with ranking citations measure (citation per paper). Since this study focused on Android malware obfuscation, we had selected 'Android malware, 'malware obfuscation', and 'malware evasion' as our search terms. The search results in 511 research from journals and conferences' proceeding database. The search results mainly records are from IEEE, journals and conferences distributions as per Table 2.

The list of collected articles represent the Android malware obfuscation and detection frameworks. It included the three types of the malware analysis techniques static, dynamic and hybrid techniques in the last decade from 2011 to early 2021. Hence, we collected Android malware frameworks for the last decade and innovative contributions that appeared in high-ranked journals or conferences such as IEEE, ACM, and Springer.

### Screening

Since, this paper explored the last 10 years' research to evaluate the Android detection frameworks against evasion techniques, we focused on experimental malware detection articles using static, dynamic and hybrid analysis techniques, excluding the unrelated articles. We excluded articles that are not Android specific malware detection such as IOS and Windows based operating system. In addition, we excluded all other languages and include only English language research to avoid translation overhead in future.

### Eligibility

As shown in Fig. 1, the review process presented four phases flow diagram, the identification collect the articles from web of science (WOS) database using above mentioned search terms, next, screening identified the criteria of article inclusion and exclusion. After removing the duplicates and excluded the non-related articles, we categorize Android malware detection by the analysis methodology static, dynamic, and

**Table 2 Comparison of the recent reviews.**

| Article type | Full name | Publisher |
|---|---|---|
| Journals | ACM Computing Surveys | ACM |
| | ACM Transaction on Computer system | ACM |
| | Computers & Security | |
| | Digital Investigation | |
| | Future Generation Computer Systems | |
| | IEEE Transactions on Dependable and Secure Computing | |
| | IEEE Access | IEEE |
| | IEEE Transactions on Industrial Informatics | IEEE |
| | IEEE Transactions on Information Forensics and Security | IEEE |
| | IEEE Transactions on Knowledge and Data Engineering | |
| | IEEE Transactions on Mobile Computing | |
| | IEEE Transactions on Network Science and Engineering | |
| | IEEE Transactions on Reliability | |
| | Information and Software Technology | |
| | Information Sciences | |
| | International Journal of Distributed Sensor Networks | |
| | International Journal of Information Security | |
| | International Journal of Interactive Multimedia & Artificial Intelligence | Springer |
| | Journal of Ambient Intelligence and Humanized Computing | |
| | Journal of artificial intelligence research | |
| | Journal of Computer Virology and Hacking Techniques | Springer |
| | Journal of Information Science and Engineering | |
| | Journal of Information Security and Applications | |
| | Journal of Supercomputing | |
| | PLOS ONE | |
| | Soft Computing | |
| | Security and Communication Networks | |
| Conferences | Advanced Computing, Networking and Security | IEEE |
| | Artificial Intelligence and Knowledge Engineering (AIKE) | IEEE |
| | Inventive Research in Computing Applications (ICIRCA) | IEEE |
| | International Arab Conference on Information Technology (ACIT) | IEEE |
| | Information Security | IEEE |
| | Network Computing and Applications (NCA) | IEEE |
| | Computer Software and Applications Conference | IEEE |
| | International Conference on Security and Privacy in Communication Systems | Springer |
| | International Conference on Security and Privacy in Communication Systems | Springer |
| | Seventh ACM on Conference on Data and Application Security and Privacy | ACM |
| | The symposium on applied computing | ACM |
| | Data and application security and privacy | ACM |

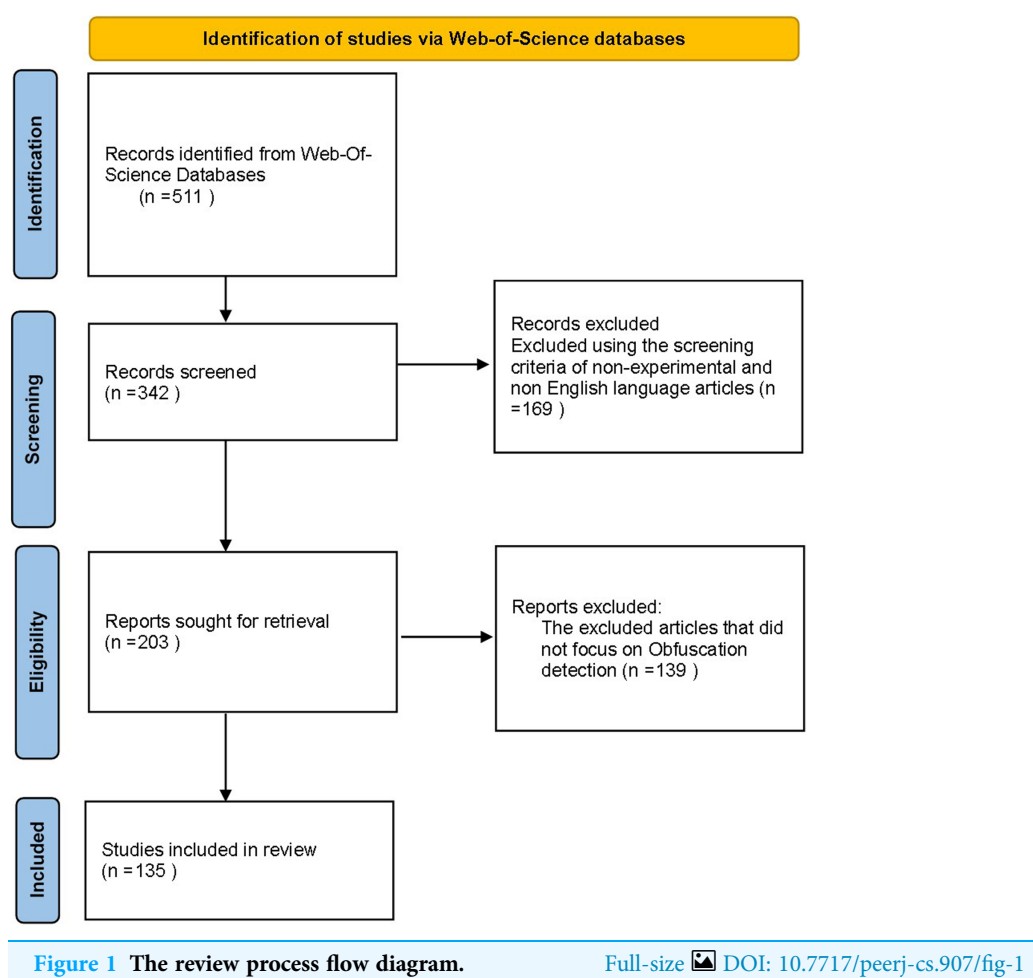

**Figure 1 The review process flow diagram.**

hybrid features. This paper decides to put metadata analysis out of this research scope. The screening phase resulted into 342 article from 511 collected in identification phase. However, we have examined 74 static analysis based frameworks. The number of dynamic based analysis frameworks are 35, the number of hybrid analysis frameworks is 26. Hence the total number of examined papers are 135 research paper that this study selected from top rank journals and conferences.

## Data analysis

We scrutinise Android malware detection academic and commercial frameworks while a large portion of the past work concentrated on commercial anti-malware solutions. This study examines different evasion techniques that hinder detecting malicious parts of applications and affect detection accuracy by reviewing state-of-the-art Android malware studies and issues limiting the detection of evasion techniques. It is worth noting that this work differs from related works that examine detection methods, as we go through evasion techniques that let malware evades detection methods.

## Android applications and weaknesses

In the section, we discussed the Android application architecture. Subsequently, we investigate the Android operating system (OS) weaknesses. This background highlights the seriousness of some drawbacks to rationalize the necessity of establishing this review and explain the essential terms to support the readers of this study.

## Android application

Android application, Android app, or APK refers to the Android application from now on and throughout this paper. APK is a compressed file; an unzipping program extracts its files and folders. This segment explains the APK components and their contents, as some terms are essential in this study. APK developers use development tools that occasionally require simple programming experience from young developers. The Android app runs on Dalvik or ART equivalent to Java Virtual Machine (JVM) in a desktop environment. The APK structure consists of many files and directories; the main file is Classes.dex Java bytecode; it includes the classes and is packed together in a single .dex file. The AndroidManifest.xml file contains deployment specifications and the required permissions from Android OS. Resources .arsc is compiled resources, and Res folder is un-compiled resources.

The Android system must install the APK file so that the end-user can utilize the application's functionalities. The Android system only accepts APK with a valid developer certificate, called developer identifier. Individual developers keep their certificate keys; there is no Central Authority (CA) server to maintain developers' keys, and thus no chain of trust between app stores and developers.

End-users need to run the installed applications, while other apps run as a service in the OS background. Therefore, the Android application's main components are as follows:

a) Activities: The user interface that end-users interact with and that communicates with other activities using intents.
b) Services: Android application component runs as a background process and bonds or un-bonds with other Android system components.
c) Broadcast and Receivers Intents: send messages that all other applications or individual applications receive.
d) Content Providers: It is the intermediate unit to share data between applications.

## Android weaknesses

With some insight into the Android applications' development design, we list the Android system's weaknesses and definitions for the readers of this study. The following is a list of Android flaws and characteristics that malware authors and attackers abuse.

(a) Open Source:

The advantage of Android source code's openness helps developers' communities enhance the OS and add more features. Therefore, the Android community improves Android OS daily. But, this contradicts with the security concerns when malware writers take this

advantage. It makes their job more straightforward than in closed source firmware, which commonly triggers new vulnerabilities and malware attacks (*Xu et al., 2016*).

(b) End-users Security Awareness:

End-users understanding malware's seriousness plays a vital role in early prevention and detection when using feedback and reviews. However, the end-users feedback system is insecure and easily polluted by fake comments (*Rashidi, Fung & Tam, 2015*). End-users click on malicious URL links in emails, web browsers, pop-ups, or Android application dialogues that download and install malicious applications. The end-users grant permissions to the apps without studying the apps' actual requirements; they believe and follow fake advertisements of permissions greedy apps.

(c) Third-party Apps Market:

Android lets end-users download applications from third-party markets and install such application offline by enabling installations from unknown sources in the phone settings menu. Several untrusted or well-verified application stores offer Android the third-party application, such as Amazon, GoApk, Slide ME, and other apps markets. In addition, there are four Chinese App markets Anzhi, Mumayi, Baidu, and eoe app third party markets, since Google Play restricted access to the Android Play Store for the Chinese population (*Fsecure, 2013*). End-users download mobile applications from any website to their mobiles devices, personal computers, or laptops *via* tools such as the ADB tool in Android SDK, which increases the probability of installing malicious apps (*Sufatrio et al., 2015b*; *Tan, Chua & Thing, 2015*).

(d) The Coarse Granularity of Android Permissions:

The Android system controls the users' application access using coarse granulated permissions, *i.e.*, one permission that provides access to entire Internet protocols and all sites. There is no competent permission administration or sufficient permission documentation, leading to excess permissions (*Fang, Han & Li, 2014*).

(e) Developers' Signatures:

Android application developers have to sign their apps with their developer key before uploading the developed application to the market. There is no external party to authenticate developers' signatures and thus no confidentiality or integrity (*Holla & Katti, 2012*). Hence, malware developers clone benign applications and sign the APK with their developer key after injecting malicious codes (*Zhang et al., 2014*). Later, malware developers upload malicious APK to third-party application markets or share the infected applications directly with their victims.

(f) Application Version Update:

Android applications usually enhance their functionalities in the form of version updates. The security frameworks analyze the application during installation, and the

update process downloads new services/features without security precautions or checks (*Luyi et al., 2014*).

(g) USB Debugging:

USB debugging is a valuable feature for Android Application development; it helps developers be more productive and efficiently troubleshoot applications. It allows direct installation of an application to the Android device using Android SDK tools such as the ADB tool. In addition Expo framework (*Zhang, Breitinger & Baggili, 2016*) has the possibility of live reloading and dynamic code loading online. On the other hand, malware writers utilize live loading features to gain remote access to install malicious applications using static and dynamic methods. The static method injects JAR (Java) or *.SO (JNI) files to the application before running, while the dynamic method call external files during runtime (*Zhang, Breitinger & Baggili, 2016*).

(h) Dynamic Code Loading (DCL):

DCL is an Android OS feature that enables benign Android applications to call another APK or malicious code to compile and execute it in real-time. However, malware developers use this feature to load their malicious codes dynamically after the detection framework ranked the malicious app as benign.

(i) Inter-application Communication (intent):

Android OS uses the inter-application intent system to deliver a message from and to applications. Malware developers sniff, modify, or gain knowledge, compromising data integrity and privacy (*Chin et al., 2011*). The intent provides flexibility in Android application development, but it is an entry point for security threats (*Feizollah et al., 2017*; *Salva & Zafimiharisoa, 2015*).

# EVASION TECHNIQUES

This section represents our taxonomy of the currently used evasion techniques and research studies on detecting obfuscated malware. Our taxonomy focuses on classifying the related studies with the same objectives and goals to harvest a comprehensive collection of material and comparative conclusions. When scrutinizing many existing studies, we find it more appropriate to study the evasion detection capabilities of each studied framework after introducing the evasion techniques that hinder malware analysis and detection. This section presents the taxonomy of detection techniques for the ground truth relation between the detection methodology and the evasion ability. Android applications have powerful tools and techniques to secure and protect their applications from being reverse-engineered. Conversely, malware authors are using obfuscation tools and techniques to evade detection. Therefore, evasions, or in other terms, transformation techniques, are techniques that try to defeat Android malware detection and rank the malware applications as benign.
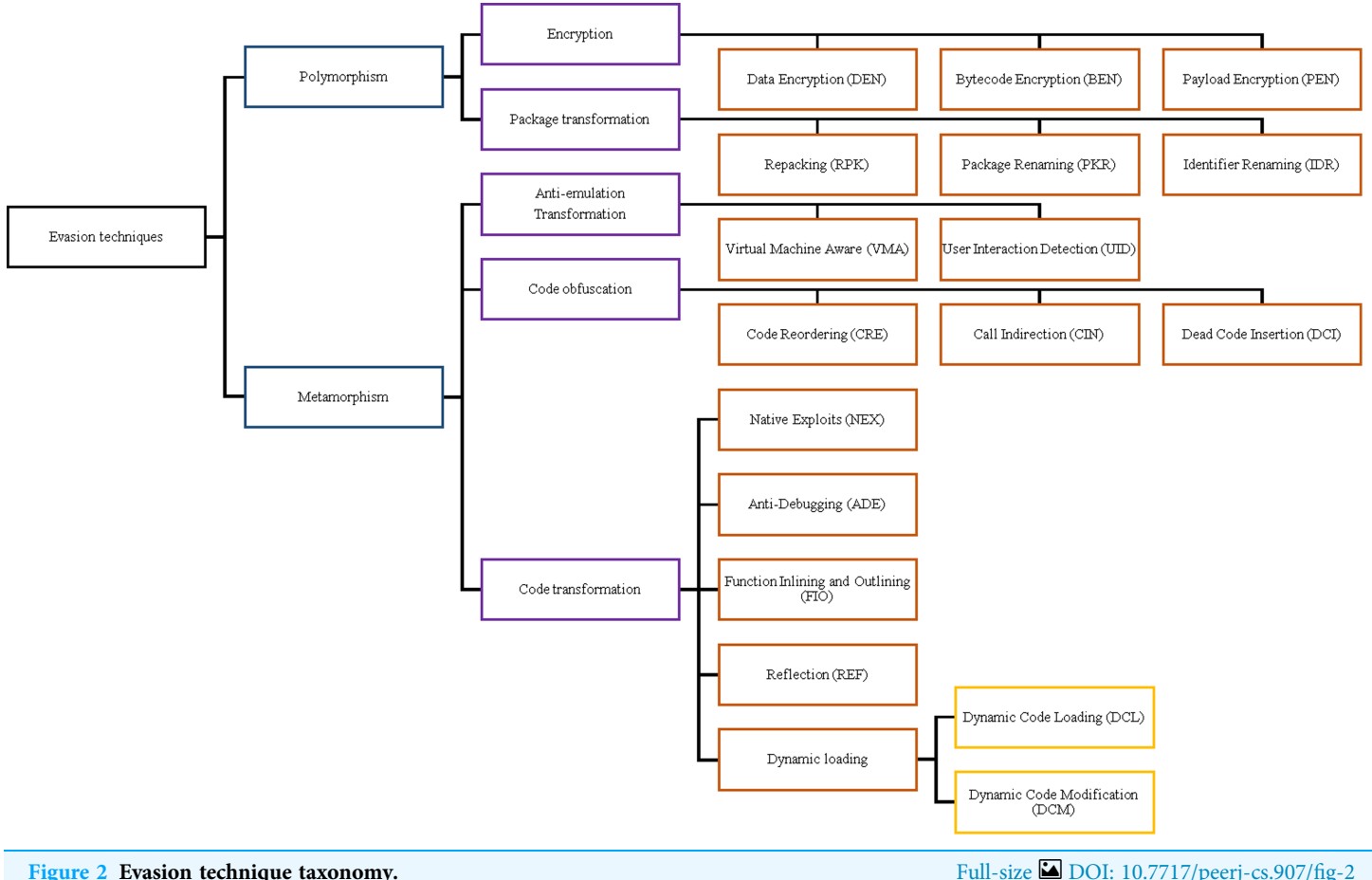

**Figure 2 Evasion technique taxonomy.**

As displayed in Fig. 2, we categorize evasion techniques into two main types. The first category is polymorphism; it transforms the malicious malware code without changing the original code of the mobile application. The second category is metamorphism, which mutates the application code, but maintains the same behaviour. Malware authors employ obfuscation tools, such as Obfuscapk (*Aonzo et al., 2020*), ProGuard (*Lafortune, 2002*), DashO (*Wang et al., 2016*), KlassMaster (*Kuhnel, Smieschek & Meyer, 2015*), and JavaGuard (*Sihag, Vardhan & Singh, 2021a*) to encrypt their code and decrypt during runtime; they modify the code itself to evade the heuristic detection and signature analysis of the malware detection techniques.

## Polymorphism

Polymorphic malware is the malware category that keeps changing its characteristics to generate different malware variants evading malware detectors. Polymorphic malware encrypts part of the code embedding malicious code. The polymorphic malwares encrypt itself with variable encryption keys but maintaining the malicious code body unaltered. Polymorphic malware is an advanced version of oligomorphic malware. The oligomorphic malware encrypts the malicious code to defeat source code static analysis based malware

detection. Usually, the malware decrypts the malware using the same techniques. However, the oligomorphic malware decrypts the encrypted malicious code using different deyrptor to make decryptor analysis more difficult. The static analysis analyze the decryptor to find the encryption key that enable the detection of the malware. Hence, the static analysis approach is not effective with oligomorphic malware. Polymorphic malware continuously change the decryptor technique to make it more difficult to the source code static analysis approach. These symptoms are indications of the presence of malicious code in an application. In this section, we discuss the polymorphism evasions subcategories, which are package transformation and encryption.

## Package transformation

In this section, we study types of package transformation, which are Repacking (RPK), Package Renaming (PKR), and Identifier Renaming (IDR).

(a) Repacking (RPK): It is the process of unpacking the APK file and repacking the original application files but signing the APK file with a developer security key (*Rastogi, Chen & Jiang, 2013*). This way, the code remains unchanged and signed the application with a different key. To repackage Android application, attackers unzips the APK file into DEX file, hence, attackers adopts reverse engineering tools to extract Java or smali code from the DEX file. Using classes, string, and methods rearrangement in DEX file, attacker modifies the architecture of the DEX arrangement resulting into defeating signature based Android malware detection. Canfora (*Canfora et al., 2015b*) considers a simple repacking evasion technique. It hinders malware detection using all of the commercial anti-malware that uses signature-based detection techniques. Thus, with every iteration, the malware's signature is changed, after which the malware can evade detection. For instance, one AnserverBot malware sample repackaged and disguised as a paid application is available on the official Android Market.

(b) Package Renaming (PKR): Every Android application has a unique package name. For instance, com.android.chrome is the package name of Google Chrome. PKR uses multilevel techniques to obfuscate the application classes except for the main Class, for instance, "FlattenPackageHireachey" or "RepackageClass" options (*Lafortune, 2002*). As shown in Algorithm 1, PKR changes all classes' names except the "MyMain" class.

This algorithm is applied relatedly to form the multilevel PKR obfuscation. The GinMaster family contains a malicious service that can root devices to escalate privileges, steal confidential information. Later, it receives instructions from a remote server to download and install applications without user interaction. The malware can successfully avoid detection by mobile anti-virus software by using polymorphic techniques to hide malicious code, obfuscating class names for each infected object, and randomizing package names and self-signed certificates for applications. Therefore, PKR evades the malware detection technique and causes false negatives, proven by *Faruki et al. (2015c)* by applying PKR to

malware applications and scanned using Virustotal platform. It shows that the repackage malware detection accuracy dropped to half in all malware categories.

(c) Identifier Renaming (IDR): Identifier is another APK parameter representing the application developer's signature. Classes, methods, and fields consider bytecode identifiers, as a signature is generated based on. Malware authors change developer identifiers using many obfuscation tools such as ProGuard (*Lafortune, 2002*) and DexGuard (*GuardSquare, 2014*) to appear as a variant application from the previously detected malicious application, leading to a different signature and evading detection methods. Real-world malware families that rename identifiers are as follows: DroidDream, Geinimi, Fakeplayer, Bgserv, BaseBridge, and Plankton.

## Encryption transformation

Some Android malware families encrypt data values inside the code, compiled code or payload, and decrypt the payload whenever desirable. This paper refers to *Data Encryption as* DEN, Bytecode Encryption as BEN, and Payload Encryption as PEN. This paper examines the following types of evasions:

a) Data Encryption (DEN): This evasion technique tends to encrypt specific data vital for the malicious action and decrypt the encrypted data later, which modifies the malware application characteristics to evade the detection techniques (*Kuhnel, Smieschek & Meyer, 2015*). The data refers to strings or network addresses embedded in the code. By encrypting such components, the malware can avoid detection methods (*Shrestha et al., 2015*), in which the authors extracted strings from APK files and analyzed the decrypted strings to detect malware. Real-world malware families that encrypt payload are as follows: DroidDream, Geinimi, Bgserv, BaseBridge, and Plankton.

b) Bytecode Encryption (BEN): using ProGuard (*Lafortune, 2002*) or DashO (*Maiorca et al., 2015*) obfuscation tools, the BEN evasion hinders reverse engineering by encrypting original code and makes it almost impossible to read. It divides the code into two parts, the encrypted and non-encrypted parts. The non-encrypted code part includes the decryption code for the encrypted part (*Faruki et al., 2014*; *Rastogi, Chen & Jiang, 2014*) during run-time. Therefore, dynamic analysis is required to detect this decryption process. However, some static analysis-based detection frameworks propose BEN evasion detection, such as DroidAPIminer (*Aafer, Du & Yin, 2013*) and Wang (*Wang & Wu, 2015*) that successfully detect BEN evasion but fail in DEN or PEN evasions detection.

c) Payload Encryption (PEN): Malware authors use payload encryption as in DroidDream (*Foremost, 2012*) malware to carry malicious payloads inside applications and install malicious applications at runtime once the system is compromised. The code is encrypted and decrypted during run time, which calls a decrypting function (*Cho, Yi & Ahn, 2018*) and runs it in real-time.

## Metamorphism

Metamorphic malware is more complex than polymorphic malware that shows a better ability to evade detection frameworks. Malware authors adopt metamorphic malware so to make metamorphic malware detection harder than leveraging polymorphic malware. The metamorphic malware writes new malicious code that varies in each iteration using the same encryption and decryption key. For example, Opcode ngrams (*Canfora et al., 2015a*) adopts the ngrams feature extraction algorithm to extract the suspected string with n count in the Opcode. It assumes that the Malware writers rarely develop metamorphic Android malware variants. Based on that assumption, it ignored the evaluation of the ngrams' detection framework against metamorphic evasions (*Canfora et al., 2015a*). Metamorphic malware rewrites itself in every iteration to evade detection methods.

## Code obfuscation

Code obfuscation is an evasion technique initially used to protect applications from piracy and illegal use by many obfuscation techniques. Conversely, malware authors use code obfuscation techniques to evade malware detections. In this study, we highlight three types of code obfuscation the *Code Reordering (CRE), Call Indirection (CIN), and Dead Code Insertion (DCI).*

a) Code Reordering (CRE): This transformation changes the order of the code by inserting the standard "goto" command to maintain the proper program instruction order.

b) Call Indirections (CIN): CIN is an object-oriented feature used dynamically to reference specific values inside the code; CIN creates code transformation evasion, obfuscating the call graph detection techniques (*Castellanos et al., 2016*; *Gascon et al., 2013*). Malware families such as DroidDream, Geinimi, and FakePlayer incorporate call indirection to evade static analysis based Android malware detection.

c) Dead Code Insertion (DCI): Malware inserts junk code into the sequence of the application to ruin its semantics. This type of transformation makes the malware more difficult to analyze (*Kwon et al., 2014*). AnDarwin (*Crussell, Gibler & Chen, 2015*) experimented with detecting Android malware based on code similarity. Their used method is unable to detect dead code insertion transformation (*Crussell, Gibler & Chen, 2015*). The code similarity approach uses a distance-vector technique, representing the distances between the original code or the DCI transformation representing a distance vector. The far the distance vector, the more complex the detection of such obfuscation.

## Advanced code transformation

This section explains the advanced code transformation techniques that are more sophisticated in hindering the malware detection frameworks. We include advanced evasion techniques, such as *Native Exploits (NEX), Function Inlining and Outlining (FIO), Reflection API (REF), Dynamic Code Loading/Modification (DCL/DCM)*, and *Anti-debugging (ADE).*

a) Native Exploits (NEX): Android applications call native libraries to run system-related functions. The malware uses a native code exploit to escalate the root privilege while running (*Xu et al., 2016*). Unfortunately, many exploits' source code is available for download. Official Android suppliers are working on a solution using regular system updates and fixes. Additionally, DroidDream malware (*Wu et al., 2015*) packs native code exploits with application payload, bypassing Android security monitoring and logging systems.

b) Function Inlining and Outlining (FIO): Inlining and outlining are compiler optimization techniques options. Inlining replaces the function call with the entire function body, and the outlining function divides the function into smaller functions. This type of transformation obfuscates the call graph detection technique by redirecting function calls and creating a maze of calls (*Gascon et al., 2013*).

c) Reflection API (REF): Reflection API is a technique to initiate a private method or get a list of parameters from another function or class, whether this class is private or public. Android developers legitimately use it to provide genericity, maintain backward compatibility, and reinforce application security. However, malware authors take advantage of this feature and use it to bypass detection methods. Reflection evasion facilitates the possibility to call private functions from any technique outside the main class. Recently few studies highlighted the reflection effect on code analysis and considered reflection during the analysis process (*Kuhnel, Smieschek & Meyer, 2015*; *Li et al., 2016*).

d) Dynamic Code Loading/Modification (DCL/DCM): Since Java has the capability of loading code at runtime using class loader methods, Android malware application dynamically download malicious code using the dynamic code loading (DCL). The DCL and DCM techniques provide advanced evasion capability to malware authors, and improper use can make benign applications vulnerable to inject malicious code. For instance, the Plankton malware family uses dynamic code loading to evade detection methods. As being the first malware with DCL that stealthy extend its capabilities on Android devices. It installs an auto-launching background application or service to the device, collecting device critical information to a server. The server sends the malicious class payload URL link to the background service using an HTTP_POST message containing a Dalvik Bytecode jar malicious payload file. In the following trigger of "init()" event of the main application, the malicious payload is invoked using the "DexClassLoader" class. Due to the unavailability of the dynamically loaded code during Android malware static analysis, the DCL and DCM evasion technique is another transformation technique that is a big challenge for static analysis (*Hsieh, Wu & Kao, 2016*; *Li et al., 2016*). Although some researchers (*Poeplau et al., 2014*; *Zhang, Luo & Yin, 2015*; *Zhauniarovich et al., 2015*) studied how DCL evades malware detection, it is still an open issue that needs more attention. Grab'n run (*Falsina et al., 2015*) uses code verification techniques to secure dynamic code loading and protect it from misuse by malware authors and attackers.

e) Anti-debugging (ADE): The malware developer presumes the limitation of Android that only one debugger can be attached to a process using ptrace functionality (*Zhang, Luo & Yin, 2015*). Hence, it prohibits attaching a debugger to the suspected application. If the malware detects the running debugging tool like Java Debug Wiring Protocol (JDWP), it discovers the operating environment running under an Android emulator or physical device. Andro-Dumpsys (*Jang et al., 2016*) is a hybrid Android malware analysis framework that claimed that it disables the attachment of "ptrace" monitoring application service to monitor the running applications, which lack ADE detection.

## Anti-emulation transformation

The primary objective of anti-emulation evasion is to detect the running environment of the sandbox and benignly masquerade as a clean application instead of launching the malicious code, which we refer to as *Virtual Machine Aware (VMA)*. Another side of anti-emulation evasion is detecting automatic user interaction emulation, which refers to as *Programmed Interaction Detection* like the monkeyrunner tool used in many frameworks, for instance, the Droidbox (*Desnos & Lantz, 2014*) sandbox tool in the Mobile-Sandbox (*Spreitzenbarth et al., 2015*).

a) Virtual Machine Aware (VMA): The dynamic analysis requires either an Android virtual machine emulator or a physical device to install the suspected application. Scientists studied the possibility of detecting the running environment fingerprints to differentiate between an emulator and a physical device (*Jing et al., 2014*; *Maier, Muller & Protsenko, 2014*; *Maier, Protsenko & Müller, 2015*; *Vidas & Christin, 2014*). Android.obad (*Faruki et al., 2015b*; *Singh, Mishra & Singh, 2015*) is an emulator-aware malware, which complicates the analysis process. The malware looks for the "Android.os.build.MODEL" value throughout the code and exits if it matches the emulator's model. The malware only runs in an emulator after patching WMA checks.

b) Programmed Interaction Detection (PID): Android malware is an event-driven application that needs a particular series of user interactions to launch malicious actions. Therefore, dynamic analysis requires a running environment user/gesture interaction. Malware writer refers to PID obfuscation as code coverage. Some researchers have tried to address code coverage; however, it remains a challenge to detect it.

We scrutinize the top Android malware detection frameworks against the two main evasion categories based on the introduced definitions of Android malware evasion techniques. The first category is polymorphism, which consists of package transformation and encryption transformation. Package transformation includes *Repacking (RPK), Package Renaming (PKR), and Identifier Renaming (IDR)*. Encryption transformation includes *Data Encryption (DEN), Bytecode Encryption (BEN), and Payload Encryption (PEN)*. The metamorphism subcategories are obfuscation transformation, advanced code

transformation, and anti-emulations transformation. The code obfuscation subcategory includes *Code Reordering (CRE), Call Indirection (CIN), and Dead Code Insertion (DCI).* Advanced code transformation includes *Native Exploits (NEX), Function Inlining and outlining (FIO), Reflection API (REF), Dynamic Code Loading/Modification (DCL/DCM), and Anti-debugging (ADE)* evasion techniques. Last but not least, anti-emulation transformation includes *Virtual Machine Aware (VMA) and Programmed Interaction Detection (PID).*

## Android evasion detection frameworks

Many researchers (*Apvrille & Apvrille, 2015*; *Bagheri et al., 2015*; *Battista et al., 2016*; *Chenxiong et al., 2015*; *Elish et al., 2015*; *Fratantonio et al., 2016*; *Gonzalez, Stakhanova & Ghorbani, 2014*; *Gurulian et al., 2016*; *Kuhnel, Smieschek & Meyer, 2015*; *Lei et al., 2015*; *Li et al., 2016*; *Martín, Menéndez & Camacho, 2016*; *Preda & Maggi, 2016*; *Sheen, Anitha & Natarajan, 2015*; *Shen et al., 2015*; *Sun, Li & Lui, 2015*; *Wang et al., 2016*; *Wu et al., 2016*; *Zhang, Breitinger & Baggili, 2016*) examine their frameworks against different evasion techniques, and they take countermeasures to overcome evasion techniques, which prevent the anti-malware framework from detecting malicious applications. These evasions are the leading cause of false negatives, as they allow many malware applications to penetrate freely into Android smart devices. This section investigates the latest frameworks with different approaches, finding a robust solution to detect evasion techniques. We are aiming to discover the gap in this area of research. We also review the different evasion test benches and tools that researchers and commercial enterprises use to secure their codes. We review the latest detection frameworks and their resilience against five different evasion categories and 16 different subcategories distributed into 56% static analysis, 28% dynamic, and 16% hybrid frameworks.

## Android malware detection techniques

There are three leading techniques for Android malware detection Fig. 3 presents the three main categories of Android malware detection techniques, the *first* category is logic-based techniques (*Lee et al., 2014*; *Zhang, She & Qian, 2015a*), based on hard-coded safe lists and predefined alarms stored in text files or a small database like Amamra (*Amamra, Robert & Talhi, 2015*). The *second* category is signature based malware detection techniques (*Niazi et al., 2015*; *Tchakounté et al., 2021*), it based the malware detection on comparing the suspicious application with malware application signature. The *third* category of Android malware detection uses machine learning (ML) classification algorithms to classify the application as benign or malware (*Afonso et al., 2015*; *Alzaylaee, Yerima & Sezer, 2016*; *Amamra, Robert & Talhi, 2015*; *Baskaran & Ralescu, 2016*; *Canfora et al., 2016*; *Canfora et al., 2015c*; *Castellanos et al., 2016*; *Faruki et al., 2015a*; *Feizollah et al., 2015*; *Fratantonio et al., 2016*; *Kurniawan, Rosmansyah & Dabarsyah, 2015*; *Lei et al., 2015*; *Lindorfer, Neugschwandtner & Platzer, 2015*; *Lopez & Cadavid, 2016*; *Meng et al., 2016*; *Nissim et al., 2016*; *Spreitzenbarth et al., 2015*; *Spreitzer et al., 2016*; *Wang & Wu, 2015*; *Wu et al., 2016*; *Xu et al., 2016*; *Yerima, Sezer & Muttik, 2014*; *Yuan, Lu & Xue, 2016*;

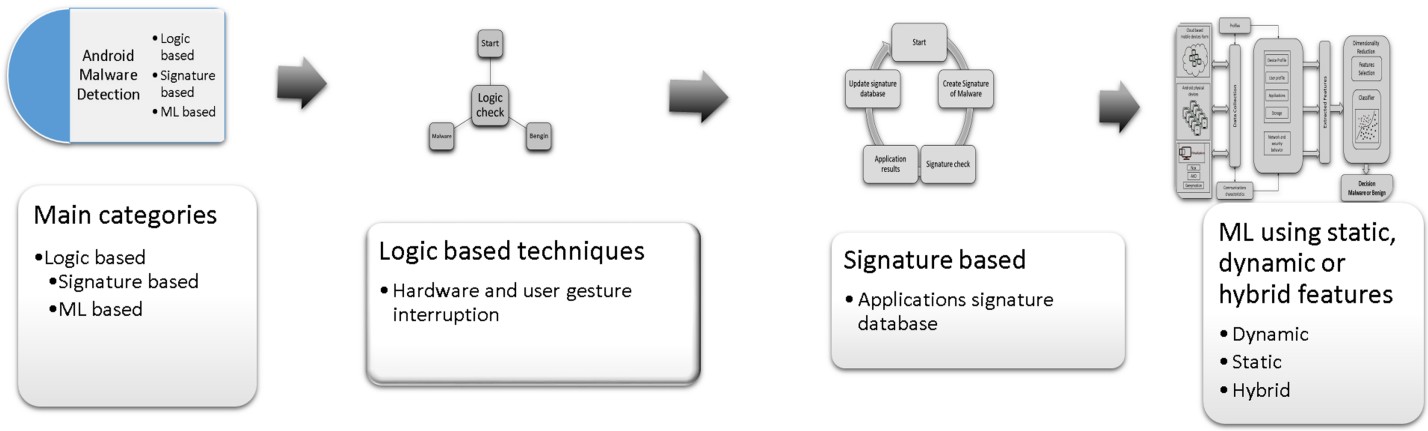

**Figure 3  The main categories of Android malware detection techniques.**

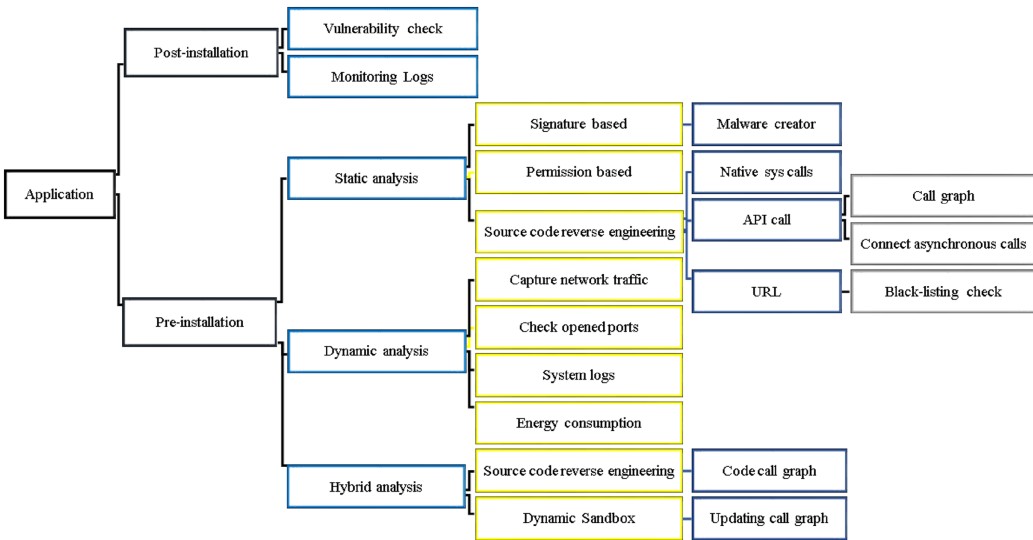

**Figure 4  Taxonomy of Android malware detection methodologies.**

*Zhang, Breitinger & Baggili, 2016*). The ML-based techniques extract the Android devices feature that represent the Android application characteristics such as the application's permission, code hierarchy from reverse engineering process, or monitoring application behaviour in runtime. The collected feature is a result of static, dynamic, or hybrid analysis of anlysing Android applications. The collected features are used to build machine learning classification model that decides whether the application is malware or benign.

Android malware detection methodologies are classified from a different point of view, as depicted in Fig. 4, defining the Android malware detection taxonomy as post-installation and pre-installation methods.

## Post-installation detection

This section explains the Android vulnerability check and monitors the system logs after installing the application. Therefore, post-installation analysis reports the security issues and malicious activity to the end-users.

a) Vulnerability Check: The vulnerability check method scans all existing Android apps and Android system versions against common security threats. APSET (*Salva & Zafimiharisoa, 2015*) collects the vulnerability pattern using the Android application's test case execution framework, which supports receiving exceptions. However, using more vulnerability patterns or generating more test cases per pattern improves the APSET malware detection performance.

b) Monitoring Logs: Android systems use process monitoring tools and network monitoring tools. Mobile-Sandbox (*Spreitzenbarth et al., 2015*) uses the process trace monitoring tool and PCAP network monitoring tool to capture the required data for analyzing the Android applications.

## Pre-installation detection

Android malware detection frameworks perform static, dynamic, or hybrid analyses to analyze features for malware detection techniques, which classify the apps as benign or malware. Hence, we identify the following application analysis methodologies.

## Static analysis

It is a technique to reverse engineer the APK statically without installing it; the analysis requires reading configuration settings, decompiling executable bytecode, and extracts the source code for further analysis.

a) Signature-based: This paper classifies the signature-based method under static analysis detection because the signature-based detection approach builds its frameworks with static Android application characteristics. As such, DroidAnalytics (*Zheng, Sun & Lui, 2013b*) uses a signature-based manner in which it dynamically collects and creates a signature for each malware and stores malware signature into a central database. This model has limitations where each of the new malware family variants needs a different signature. LimonDroid (*Tchakounté et al., 2021*) proposed a signature-based database of Android malware signature based on fuzzy hashing technique. It builds a signature database for literature purposes rather than a malware detection framework.

b) Permission-based: APK Auditor (*Talha, Alper & Aydin, 2015*) is a static model that leverages *permission-based* detection castoff decompressing the APK package; it extracts the malicious symptoms using permission and signature matching analysis. Likewise, Triggerscope (*Fratantonio et al., 2016*) uses permissions characteristics as an input to classify the application using different machine learning algorithms (*Abdulla & Altaher, 2015*; *Alazab et al., 2020*; *Arora, Peddoju & Conti, 2019*; *Dharmalingam & Palanisamy, 2021*; *Fang, Han & Li, 2014*; *Glodek & Harang, 2013*;

*Li et al., 2018*; *Niazi et al., 2015*; *Şahin et al., 2021*; *Shalaginov & Franke, 2014*; *Talha, Alper & Aydin, 2015*; *Tiwari & Shukla, 2018*).

c) Source code based Analysis: *Arp et al. (2015)* extracts features from the application's *Androidmanifest file* and *source code*; it scrutinizes the code by listing the *native calls*, *API calls*, and *URL addresses*. It uses machine learning classification to discriminate between malware and benign apps. Likewise, DroidMat (*Wu et al., 2012*) uses the configuration file to get the required permission by the APK and counts the method that has API calls from the decompiled source code; it uses 1,500 benign APK applications and 238 malware, evaluates the accuracy of the framework, and achieves 97.87% accuracy. However, *Lei et al. (2015)* proposed a probabilistic discriminative model based on decompiled source code with permissions. It classified apps as benign and malware using machine learning classification techniques. *Hanna et al. (2013)* tried to find the code similarity among Android applications to detect similar code patterns with the same vulnerabilities and the repackaged or cloned applications in Android markets.

## Dynamic analysis

Dynamic analysis is the process of running the suspect app in an isolated Android environment. It starts by receiving the Android application APK files, either using an online scanning portal VirusTotal (Google) or a scanning agent on an Android smartphone/device. Next is opening a suitable Android operating environment in a physical device or emulator, which we hereafter refer to as a sandbox. The sandbox isolates the application to protect the analysis device from possible malicious attacks. Later, the dynamic analysis starts system logging and network monitoring tools and captures the default system logs.

Once the sandbox and the logging or monitoring tools are ready, the APK installation follows, and once the installation is successful, the logging system captures all system logs. Dynamic analysis requires the application to start and run all codes and capture all changes to the Android system environment. The sandbox captures the system logs before installing the application and compares the system logs after installing and running the suspect Android application. The sandbox uses a monkeyrunner tool to randomly emulate user gestures and cover all the possible alleged code in an Android application. Dynamic analysis sandboxing techniques install and run Android applications in a virtual environment, emulator, or physical device and monitor the application's behaviour. It considers network traffic, opened ports, and system calls. One of the main issues during the monitoring process is the user interaction simulation tool, which simulates the user interaction gestures that must cover all possible interactions. The following are types of sandboxing: *Sandbox Emulator*: Most researchers (*Afonso et al., 2015*; *Desnos & Lantz, 2014*; *Faruki et al., 2015a*; *Spreitzenbarth et al., 2015*) use Android emulators like Droidbox (*Desnos & Lantz, 2014*), TantDroid (*Chao et al., 2020*), and CuckooDroid (*Check Point Software Technologies, 2015*), which run an Android image as a virtual machine. Later, the framework destroys the used OS image and prepares a factory reset Android OS for the

following analysis process. *Physical sandbox device*: The dynamic analysis algorithm resets the physical device to factory settings to make sure the analysis captures only the suspected application's behaviour. It overcomes the limitations of using emulators and uses physical devices to analyze suspicious applications (*Shrestha et al., 2015*) dynamically.

Android malware dynamic analysis faces some challenges; some malware families evade the dynamic malware analysis environment by halting the malicious download until the dynamic analysis finishes the monitoring period. The sandbox environment suffers from the computational time required to load the Android operating system, create log files, install APK, capture system logs and network traffic, and copy the log files to form understandable characteristics. User gestures emulation using Android tools, such as monkeyrunner, is less precise and partially covers the code of an application. Phone calls, SMS, GPS, and NFC hardware emulation is another challenge in Android malware dynamic analysis, as they are not as realistic as a physical device. The dynamic analysis kills the emulator after the dynamic analysis time. Therefore, the dynamic analysis launches a new emulator instance needs for every App analysis. These challenges prevent the dynamic analysis from performing effective malware detection. Some studies have considered dynamic analysis to overcome the limitations of static analysis (*Afonso et al., 2015*; *Amos, Turner & White, 2013*; *Desnos & Lantz, 2014*; *Enck et al., 2014a*, *2014b*; *Lindorfer, Neugschwandtner & Platzer, 2015*; *Spreitzenbarth et al., 2015*; *Wang & Shieh, 2015*; *Zhao et al., 2014*).

## Hybrid analysis

The hybrid-based detection frameworks, like Mobile-Sandbox (*Spreitzenbarth et al., 2015*), Droiddetector (*Yuan, Lu & Xue, 2016*), and Andro-Dumpsys (*Jang et al., 2016*), combine the dynamic analysis and static analysis techniques to reconcile the limitations of the static analysis. The hybrid analysis extracts static features using reverse engineering techniques (*Lim et al., 2016*). Static features are apps permissions, code analysis, intent, network address, string, and hardware features. Likewise, it extracts the dynamic analysis of the application by capturing the network traffic, system calls, user interaction, and system components using sandbox methodologies. Later, it combines a group of static and dynamic features, driving the machine learning algorithms to classify the application to benign or malware.

## Android malware dataset

Most Android malware detection frameworks adopt machine learning algorithms to build a detection model; hence researchers crawl apps from the official apps market store Google Play to build its dataset (*Arp et al., 2015*; *Parkour, 2013*; *Yajin & Xuxian, 2012*). It also crawls sample applications from third-party application stores, such as Soc.io Mall, Samsung Galaxy apps, SlideME, AppsLib, GetJar, Mobango, Opera Mobile Store, Amazon Appstore, and 1Mobile markets. To label the crawled applications as benign or malware, researchers employ online security scanning tools as listed in Table 3. For instance, Virustotal and AndroTotal, and the online service are used to scan the crawled apps and cluster the found malware apps into malware families. Researchers label all crawled apps

**Table 3 Online malware scanning frameworks.**

| Online security scanning | Description | Started | Scanning rate (app/day) | Services | License |
|---|---|---|---|---|---|
| VirusTotal (*Google, 2011*) | https://www.virustotal.com | 2011 | Ignored | Web/API | Free |
| AndroTotal (*Maggi, Valdi & Zanero, 2013*) Droydseuss (*Coletta, Van der Veen & Maggi, 2016*) | https://andrototal.org/ http://droydseuss.com | 2013 | Ignored | Web | Free |
| ANDRUBIS (*Lindorfer et al., 2014*) | https://anubis.iseclab.org commercialized to https://www.lastline.com/ | 2012 | 3,500 | API | Free/discontinued– Paid only |
| APK Auditor (*Talha, Alper & Aydin, 2015*) | http://app.ibu.edu.tr:8080/apkinspectoradmin | 2015 | Ignored | Web | Discontinued |
| NVISO (*Hoffmann et al., 2016*) | https://apkscan.nviso.be/ | – | 2,400 | Web/API | Free/Pro |
| Copperdroid | http://copperdroid.isg.rhul.ac.uk/copperdroid/ | 2015 | NA | Web | NA |
| Totalhash | https://totalhash.cymru.com | | 10 | Web/API | Commercial |

using VirusTotal to build Android malware detection datasets. Many of the dataset are published for future academic research such as Drebin (*Arp et al., 2015*), Genome (*Yajin & Xuxian, 2012*), Kharon (*Kiss et al., 2016*), AMD (*Li et al., 2017*), AAGM (*Lashkari et al., 2017*), PRAGuard (*Maiorca et al., 2015*), AndroZoo (*Allix et al., 2016*) datasets.

## Machine learning in android malware detection

Based on collected characteristics or so-called features (*Feizollah et al., 2015*), different machine learning classification techniques classify APK as benign or malware. However, deep insight into machine learning techniques is outside the scope of this study. Android malware detection classifies Android apps into two classes benign and malware. However, some papers detect Android Ransomware (*Andronio, Zanero & Maggi, 2015*; *Maiorca et al., 2017*) considering three classes benign, malware, and ransomware. Hence, we briefly explain the evaluation measures of ML classification. Machine learning comprises three main categories, namely supervised, unsupervised, and reinforcement learning.

(a) Supervised Model:

Supervised machine learning bases its model on a labelled dataset. The framework splits the dataset into two subsets; first subset is for training and creating the classification model, and the second subset is for testing and validating the trained classification model. Most researchers split the data into 70% training and 30% testing subsets, but some split the data into 50% for training and 50% for testing (*Adebayo & AbdulAziz, 2014*).

(b) Unsupervised Model:

In the unsupervised model, apps are unlabeled. The unsupervised model recognizes the class of the applications without knowing which App is malware or benign. Researchers use unsupervised models to learn the covert pattern of the unlabeled data

**Table 4 Confusion matrix.**

| | | Classified apps | |
| | Total samples | Malware | Benign |
|---|---|---|---|
| True apps | Malware - M | TP[1] | FN[2] |
| | Benign - B | FP[3] | TN[4] |

Notes:
[1] TP True Positive.
[2] FN False Negative.
[3] FP False Positive.
[4] TN True Negative.

(*Akpojaro, Aigbe & Onwudebelu, 2014*; *Kohout & Pevny, 2015*; *Tang, Sethumadhavan & Stolfo, 2014*).

(c) Reinforcement Learning:

The machine exposes itself to an environment where it trains itself continually using trial and error. This machine learns from experience and tries to capture the best possible knowledge to make accurate business decisions. An example of reinforcement learning is the Markov Decision Process (*Kaelbling, Littman & Moore, 1996*).

To understand the supervised model classification performance, ML introduces the confusion matrix to calculate the performance measures as per Table 4. Let D be the total number of test apps, which we use to examine the supervised ML model performance that classifies apps as benign or malware, let M be the number of malware samples, and B the number of benign samples.

True Positive (TP) represents the number of malware correctly classified.

False Positive (FP) accounts for the number of benign apps classified erroneously as malware.

True Negative (TN) represents the number of correctly classified benign apps.

False Negative (FN) accounts for the number of malware apps classified erroneously as benign.

The ML performance measures represent the accuracy of the Android malware detection classification frameworks. Table 5 explains the ML performance measure formulas and their direct mathematical relation to the confusion matrix.

The Receiver Operating Characteristic (ROC) curve plots the TPR against FPR where TRP is the *y*-axis and FPR is the *x*-axis. Every point in the ROC curve represents one confusion; it is all based on TP and FP values. Area Under the Curve (AUC) is the area under the ROC curve representing the aggregation of the ML trained model (*Afifi et al., 2016*; *Baskaran & Ralescu, 2016*; *Feizollah et al., 2015*).

## Evasion test benches tools

Researchers or commercial companies have developed the evasion test benches to study the robustness of the currently available anti-malware applications or protect their software packages from piracy issues. The first test benches trials were ADAM

**Table 5  ML classification performance measures.**

| Performance measure | Short-form | Formulas | Description |
|---|---|---|---|
| Recall or Sensitivity | TPR | $= \dfrac{TP}{M} = \dfrac{TP}{TP+FP}$ | True Positive Rate |
| Miss rate | FNR | $= \dfrac{FN}{M} = \dfrac{FN}{TP+FP}$ | False Negative Rate |
| Fall-out | FPR | $= \dfrac{FP}{B} = \dfrac{FP}{TP+FN}$ | False Positive Rate |
| Specificity | TNR | $= \dfrac{TN}{B} = \dfrac{TN}{TP+FN}$ | True Negative Rate |
| Precision | PPV | $= \dfrac{TP}{TP+FP}$ | Positive Predictive Value |
| False Discovery Rate | FDR | $= \dfrac{FP}{TP+FP}$ | False Discovery Rate |
| False Omission Rate | FOR | $= \dfrac{FN}{TN+FN}$ | False Omission Rate |
| Negative Predictive Value | NPV | $= \dfrac{TN}{TN+FN}$ | Negative Predictive Value |
| Accuracy | ACC | $= \dfrac{TP+TN}{D} = \dfrac{TP+TN}{TP+TN+FP+FN}$ | Total truly detected apps over total examined apps |
| F-measure | F1 | $= \dfrac{2 \times TP}{2 \times TP + FN + FP}$ | The harmonic mean of precision and sensitivity |

(*Zheng, Lee & Lui, 2013a*) and Droidchameleon Rastogi (*Rastogi, Chen & Jiang, 2013*), which conclude that there is a detection performance degradation when applying trivial obfuscation techniques. However, researchers developed evasions tools to evaluate commercial anti-malware performance, such as PANDORA (*Protsenko & Muller, 2013*), Mystique (*Meng et al., 2016*), AAMO (*Preda & Maggi, 2016*), ProGuard (*Lafortune, 2002*), and others as listed in Table 6. Evasion tools were initially aiming to protect commercial software companies' applications from piracy, such as DexGuard (*GuardSquare, 2014*), which is an extension of ProGuard (*Lafortune, 2002*), and Klassmaster (*Klassmaster, 2013*). Recently, a pretty good number of researchers develop frameworks targeting obfuscation and malware variant resiliency. PetaDroid (*Karbab & Debbabi, 2021*) introduces the severe first obfuscation dataset, which is a good initial. However, it proves that the accuracy degrades with time and needs malware variant and obfuscation adaptations. Dynamic analysis frameworks (*Chen et al., 2018*; *Cho, Yi & Ahn, 2018*; *De Lorenzo et al., 2020*; *Feng et al., 2018*; *Sihag et al., 2021*; *Xue et al., 2017*) declare the ability to detect all types of obfuscated malware; however, most of it misses the evaluation report of each obfuscation technique using obfuscated malware datasets. Researchers who evaluated their framework against particular evasions are identified by mentioning the detected evasion, which represents that the respective study either evaluated or presumed its ability to detect the evasion technique, while "Failed to detected or ignored" means the respective study is defeated the corresponding evasion technique. The "stared" cell indicates the framework that ignores the evaluation experiments on evasion techniques or assumptions to that effect, or the study misses evaluating its framework performance against this evasion technique.

**Table 6 Android malware evasion test benches.**

| Framework | Polymorphism | | | | | | Metamorphism | | | | | | | | | |
|---|---|---|---|---|---|---|---|---|---|---|---|---|---|---|---|---|
| | Package transformation | | | Encryption | | | Code obfuscation | | | Advanced code transformation | | | | | Anti-emulator | |
| | (RPK) | (PKR) | (IDR) | (DEN) | (BEN) | (PEN) | (CRE) | (CIN) | (DCI) | (NEX) | (FIO) | (REF) | DCL/DCM | (ADE) | (VMA) | (PID) |
| ADAM (*Zheng, Lee & Lui, 2013a*) | ✓ | * | * | ✓ | * | * | ✓ | * | ✓ | * | * | * | * | * | * | * |
| DroidChameleon (*Rastogi, Chen & Jiang, 2013*) | ✓ | * | * | * | * | * | ✓ | * | * | * | * | ✓ | * | * | * | * |
| ProGuard (*Lafortune, 2002*) | * | * | * | ✓ | ✓ | ✓ | * | * | * | * | * | * | * | * | * | * |
| DexGuard (*GuardSquare, 2014*) | * | * | * | ✓ | * | * | ✓ | ✓ | * | * | * | * | * | * | * | * |
| Klassmaster (*Klassmaster, 2013*) | * | * | * | ✓ | ✓ | * | ✓ | ✓ | * | * | * | * | * | * | * | * |
| Maiorca (*Maiorca et al., 2015*) | ✓ | * | * | ✓ | ✓ | ✓ | * | * | * | * | * | ✓ | * | * | * | * |
| Vidas (*Vidas & Christin, 2014*) | * | * | * | * | * | * | * | * | * | * | * | * | * | * | ✓ | * |
| Petsas (*Petsas et al., 2014*) | * | * | * | * | * | * | * | * | * | * | * | * | * | * | ✓ | * |
| Morpheus (*Jing et al., 2014*) | * | * | * | * | * | * | * | * | * | * | * | * | * | * | ✓ | * |
| Garcia (*Garcia et al., 2015*) | * | ✓ | * | ✓ | ✓ | * | * | ✓ | * | * | * | * | * | * | * | * |
| DroidSieve (*Suarez-Tangil et al., 2017*) | * | * | * | ✓ | ✓ | ✓ | * | * | * | * | * | ✓ | ✓ | * | * | * |
| MysteryChecker (*Jeong et al., 2014*) | ✓ | * | * | * | ✓ | ✓ | ✓ | ✓ | * | * | * | * | * | * | * | * |
| PANDORA (*Protsenko & Muller, 2013*) | * | * | * | ✓ | * | * | * | * | * | * | ✓ | ✓ | * | * | * | * |
| Mystique (*Meng et al., 2016*) | * | * | ✓ | ✓ | * | * | * | * | * | * | ✓ | * | * | * | * | * |
| Canfora (*Canfora et al., 2015b*) | ✓ | ✓ | ✓ | ✓ | * | * | ✓ | * | ✓ | * | * | * | * | * | * | * |
| Hatwar (*Hatwar & Shelke, 2014*) | * | * | * | * | * | * | * | * | * | * | * | * | ✓ | * | * | * |
| AAMO (*Preda & Maggi, 2016*) | ✓ | ✓ | * | * | ✓ | * | ✓ | ✓ | ✓ | * | ✓ | ✓ | * | ✓ | * | * |

(Continued)

| Framework | Polymorphism | | | | | | Metamorphism | | | | | | | | Anti-emulator | |
| | Package transformation | | | Encryption | | | Code obfuscation | | | Advanced code transformation | | | | | | |
| | (RPK) | (PKR) | (IDR) | (DEN) | (BEN) | (PEN) | (CRE) | (CIN) | (DCI) | (NEX) | (FIO) | (REF) | DCL/DCM | (ADE) | (VMA) | (PID) |
|---|---|---|---|---|---|---|---|---|---|---|---|---|---|---|---|---|
| Abid (*Abaid, Kaafar & Jha, 2017*) | * | * | * | * | * | * | * | * | * | * | * | * | ✓ | * | * | * |
| EnDroid (*Feng et al., 2018*) | * | * | * | * | * | * | * | * | * | * | * | ✓ | ✓ | * | * | * |
| Bacci (*Bacci et al., 2018*) | ✓ | ✓ | ✓ | ✓ | * | * | ✓ | ✓ | ✓ | * | * | * | * | * | * | * |
| DexMoinitor (*Cho, Yi & Ahn, 2018*) | * | * | * | ✓ | ✓ | ✓ | * | * | * | * | * | * | * | * | * | * |
| Kim (*Kim et al., 2019*) | * | ✓ | ✓ | ✓ | * | * | * | ✓ | ✓ | * | * | * | * | * | * | * |
| DAMBA (*Zhang et al., 2020*) | * | * | * | ✓ | ✓ | ✓ | * | ✓ | * | * | * | * | ✓ | * | * | * |
| IMCFN (*Vasan et al., 2020*) | ✓ | ✓ | ✓ | ✓ | * | * | ✓ | * | ✓ | * | * | * | * | * | * | * |
| PetaDroid (*Karbab & Debbabi, 2021*) | ✓ | ✓ | ✓ | ✓ | * | ✓ | ✓ | ✓ | ✓ | * | * | ✓ | * | * | * | * |
| BLADE (*Sihag, Vardhan & Singh, 2021b*) | ✓ | ✓ | ✓ | ✓ | ✓ | ✓ | ✓ | * | * | * | * | * | * | * | * | * |
| DANDroid (*Millar et al., 2020*) | * | * | * | ✓ | ✓ | ✓ | * | * | * | * | * | * | * | * | * | * |
| AndrODet (*Mirzaei et al., 2019*) | ✓ | ✓ | ✓ | ✓ | * | * | * | ✓ | * | * | * | * | * | * | * | * |
| Dadidroid (*Ikram, Beaume & Kâafar, 2019*) | ✓ | ✓ | ✓ | ✓ | ✓ | ✓ | * | ✓ | * | * | * | * | * | * | * | * |
| Obfusifier (*Li et al., 2019*) | ✓ | ✓ | ✓ | * | * | * | ✓ | ✓ | ✓ | * | * | * | * | * | * | * |

**Note:**
RPK, Repacking; PKR, Package Renaming; IDR, Identifier Renaming; DEN, Data Encryption; BEN, Bytecode Encryption; PEN, Payload Encryption; CRE, Code Reordering; CIN, Call Indirections; DCI, Dead Code Insertion; NEX, Native Exploits; FIO, Function Inlining and Outlining; API (REF), Reflection; DCL/DCM, Dynamic code loading/Modification; ADE, Anti-debugging; VMA, Virtual Machine Aware; PID, Programmed Interaction Detection.

# EVALUATION OF EVASION DETECTION FRAMEWORKS

We have explored the last 10 years' research to evaluate the Android detection frameworks against evasion techniques discussed in evasion techniques section. We studied Android malware detection frameworks for the last decade from 2011 to early 2021, as listed in Table 7. We categorize malware detection framework by the analysis methodology static,

**Table 7 List of examined Android malware detection frameworks.**

| Detection techniques | The examined Android malware detection frameworks | Number of frameworks |
|---|---|---|
| Static | DroidMat (*Wu et al., 2012*), Juxtapp (*Hanna et al., 2013*), DroidOLytics (*Faruki et al., 2013*), Zhou (*Zhou et al., 2013*), DroidAPIMiner (*Aafer, Du & Yin, 2013*), MAMA (*Sanz et al., 2013*), QuantDroid (*Markmann, Gessner & Westhoff, 2013*), Glodek (*Glodek & Harang, 2013*), ViewDroid (*Zhang et al., 2014*), Yerima (*Yerima, Sezer & Muttik, 2014*), DroidGraph (*Kwon et al., 2014*), MysteryChecker (*Jeong et al., 2014*), AdDetect (*Narayanan, Chen & Chan, 2014*), ResDroid (*Shao et al., 2014*), Dendroid (*Suarez-Tangil et al., 2014*), Wei et al. (*Wei et al., 2015*), Poeplau (*Poeplau et al., 2014*), Chen (*Chen et al., 2015*), Apk Auditor (*Talha, Alper & Aydin, 2015*), Abdulla (*Abdulla & Altaher, 2015*), Andro-Tracer (*Kang et al., 2015*), Dempster–Shafe (*Du, Wang & Wang, 2015*), Dexhunter (*Zhang, Luo & Yin, 2015*), DroidExec (*Wei et al., 2015*), AnDarwin and DNADroid (*Crussell, Gibler & Chen, 2015*), AndroSimilar (*Faruki et al., 2015d*), Grab 'n Run Falsina (*Falsina et al., 2015*), Ngrams (*Canfora et al., 2015a*), SeqMalSpec -Sufatrio (*Sufatrio et al., 2015a*), DroidEagle (*Sun, Li & Lui, 2015*), VulHunter (*Chenxiong et al., 2015*), COVERT (*Bagheri et al., 2015*), Sheen (*Sheen, Anitha & Natarajan, 2015*), Droidkin (*Gonzalez, Stakhanova & Ghorbani, 2014*), Shen (*Shen et al., 2015*), SherlockDroid (*Apvrille & Apvrille, 2015*), Kuhnel (*Kuhnel, Smieschek & Meyer, 2015*), Elish (*Elish et al., 2015*), Lei (*Lei et al., 2015*), Gurulian (*Gurulian et al., 2016*), TriggerScope (*Fratantonio et al., 2016*), Wu (*Wu et al., 2016*), DroidRA (*Li et al., 2016*), AAMO (*Preda & Maggi, 2016*), Wang (*Wang et al., 2016*), MocDroid (*Martín, Menéndez & Camacho, 2016*), Battista (*Battista et al., 2016*), RAPID Zhang (*Zhang, Breitinger & Baggili, 2016*), DroidSieve (*Suarez-Tangil et al., 2017*), Bhandari et al., (*Bhandari et al., 2017*), Jin Li (*Li et al., 2018*), AndRODet (*Mirzaei et al., 2019*), PetaDroid (*Karbab & Debbabi, 2021*), Amin (*Amin et al., 2020*), Taheri (*Taheri et al., 2020*), ProDroid (*Sasidharan & Thomas, 2021*), Tiwari (*Tiwari & Shukla, 2018*), GDroid (*Gao, Cheng & Zhang, 2021*), Millar (*Millar et al., 2021*), Şahin (*Şahin et al., 2021*), DGCNDroid (*Yang et al., 2021*), IntDroid (*Zou et al., 2021*), Dharmalingam (*Dharmalingam & Palanisamy, 2021*), BLADE (*Sihag, Vardhan & Singh, 2021b*), Wang (*Wang et al., 2020*), Pektas (*Pektaş & Acarman, 2020*), Alazab (*Alazab et al., 2020*), Jung (*Jung et al., 2018*), Tiwari (*Tiwari & Shukla, 2018*), Maiorca (*Maiorca et al., 2017*), Alahy (*Alahy et al., 2020*), Hamming (*Taheri et al., 2020*), SEDMDroid (*Zhu et al., 2020*), Kim Multimodal (*Kim et al., 2019*), Taha (*Taha & Malebary, 2021*), Dadidroid (*Ikram, Beaume & Kâafar, 2019*), Obfusifier (*Li et al., 2019*) | 74 |
| Dynamic | Amos (*Amos, Turner & White, 2013*), AndroTotal (*Maggi, Valdi & Zanero, 2013*), Lee & Kim (*Lee et al., 2014*), TaintDroid (*Enck et al., 2014a*), Pektas (*Pektas & Acarman, 2014*), Soh (*Soh et al., 2015*), Shabtai (*Shabtai et al., 2014*), VetDroid (*Yuan et al., 2014b*), DroidBarrier (*Almohri, Yao & Kafura, 2014*), APSET (*Salva & Zafimiharisoa, 2015*), Afonso (*Afonso et al., 2015*), Maier (*Maier, Protsenko & Müller, 2015*), Singh (*Singh, Mishra & Singh, 2015*), Gheorghe (*Gheorghe et al., 2015*), DwroidDump (*Kim, Kwak & Ryou, 2015*), Ng (*Ng & Hwang, 2015*), GroddDroid (*Abraham et al., 2015*), Wu (*Wu et al., 2015*), DynaLog (*Alzaylaee, Yerima & Sezer, 2016*), Q-floid (*Castellanos et al., 2016*), Diao (*Diao et al., 2016*), Alzaylaee (*Alzaylaee, Yerima & Sezer, 2017*), (*Feng et al., 2018*), DE-LADY (*Sihag et al., 2021*), Wang (*Wang & Li, 2021*), MLDroid (*Mahindru & Sangal, 2021*), Liu (*Liu et al., 2021*), BPFroid (*Agman & Hendler, 2021*), DL-Droid (*Alzaylaee, Yerima & Sezer, 2020*), Droidetec (*Ma et al., 2020*), Taheri (*Taheri et al., 2020*), Abuthawabeh (*Abuthawabeh & Mahmoud, 2019*), Feng (*Feng et al., 2020*), Wang (*Wang et al., 2019*), Chen (*Chen et al., 2018*) | 35 |
| Hybrid | RiskRanker (*Grace et al., 2012*), MobSafe (*Xu et al., 2013*), Shalaginov (*Shalaginov & Franke, 2014*), ARIGUMA (*Zhong et al., 2013*), Petsas (*Petsas et al., 2014*), Droid-Sec (*Yuan et al., 2014a*), AMDetector (*Zhao et al., 2014*), MARVIN (*Lindorfer, Neugschwandtner & Platzer, 2015*), Mobile-Sandbox (*Spreitzenbarth et al., 2015*), StaDyna (*Zhauniarovich et al., 2015*), Tap-Wave-Rub (*Shrestha et al., 2015*), Droiddetector (*Yuan, Lu & Xue, 2016*), Andro-Dumpsys (*Jang et al., 2016*), Abaid (*Abaid, Kaafar & Jha, 2017*), Manto (*Mantoo & Khurana, 2020*), Chao (*Chao et al., 2020*), Loreenzo (*De Lorenzo et al., 2020*), Puerta (*de la Puerta et al., 2019*), Surendrean (*Surendran, Thomas & Emmanuel, 2020*), Lu (*Lu et al., 2020*), Dhalaria (*Dhalaria & Gandotra, 2021*), Zhu (*Zhu et al., 2021*), Nawaz (*Nawaz, 2021*), Liu (*Liu et al., 2021*), PNSDroid (*Kandukuru & Sharma, 2018*), Bacci (*Bacci et al., 2018*), DAMBA (*Zhang et al., 2020*) | 26 |

dynamic, and hybrid features. This paper decides to put metadata analysis out of this research scope. We have examined 74 static analysis based frameworks. The number of dynamic based analysis frameworks are 35. The number of hybrid analysis frameworks is

26. Hence, the total number of examined papers are 135 research paper that this study selected from top rank journals and conferences.

## Polymorphism evasion detection

We examine the three main static, dynamic, and hybrid frameworks *vs* polymorphism evasions. Table 8 represents static, dynamic, and hybrid analysis based detection; we scrutinize each framework against polymorphism transformation techniques in the two categories package transformation and encryption transformation. Each framework uses various samples of Android malware and benign applications' datasets in the evaluation process; each dataset contains a certain number of malware and benign applications. For instance, APK Auditor (*Talha, Alper & Aydin, 2015*) tested its framework against 6,909 malware and 1,853 benign applications; a total of 8,762 apps that APK Auditor crawled from Google play store and other datasets such as Genome Project and Contagio. APK Auditor achieved 88% malware detection accuracy. As it is signature-based, most of the evasion techniques prevent the APK Auditor detection framework from detecting malware applications.

(a) Package Transformation:

– RPK - Repacking Evasion Detection:

Detecting repacking evasion is possible using static analysis and detection techniques; Dempster–Shafe (*Du, Wang & Wang, 2015*) investigate repacking characteristics using a control flow graph and claimed better resistance to code obfuscation techniques. Likewise, Droidgraph (*Kwon et al., 2014*) used the hierarchical class levels to determine the repackaged malicious code to the original payload; it also considered the API calls, junk code, and code obfuscation. It reduced the code comparison time compared to the polynomial time-consuming native call graphs algorithm. Though, reflection successfully evades the detection framework that uses the control flow graph. Other static detection approaches such as MysteryChecker (*Jeong et al., 2014*), AnDarwin (*Crussell, Gibler & Chen, 2015*), AndroSimilar (*Faruki et al., 2015d*), ngrams (*Canfora et al., 2015a*), DroidEagle (*Sun, Li & Lui, 2015*), DroidKin (*Gonzalez, Stakhanova & Ghorbani, 2014*), DroidOlytics (*Faruki et al., 2013*), Gurulian (*Gurulian et al., 2016*), Shen (*Shen et al., 2015*), and AAMO (*Preda & Maggi, 2016*) have indicated their ability to detect RPK evasions. While studying dynamic analysis papers, we notice that most dynamic studies provide less attention to this evasion type. Similarly, *Soh et al. (2015)* and *Wu et al. (2015)* stressed that RPK evasion detection could detect RPK evasion, as illustrated in Table 8. The study spotted 20 papers that scrutinized the RPK evasion using static analysis, and only two papers scrutinized RPK using dynamic analysis.

– PKR - Package Renaming Detection:

Static analysis frameworks such as DroidoLytics (*Faruki et al., 2013*) and Droidkin (*Gonzalez, Stakhanova & Ghorbani, 2014*) examine their capability in detecting PKR evasion techniques. However, many other papers insufficiently evaluate its framework

against PKR, such as APK Auditor (*Talha, Alper & Aydin, 2015*), DroidGraph (*Kwon et al., 2014*), Andro-tracer (*Kang et al., 2015*), Vulhunter (*Chenxiong et al., 2015*), and COVERT (*Bagheri et al., 2015*), as presented in Table 8. Dynamic and Hybrid analysis frameworks studies incompetently examine its robustness against PKR, except one research, Shen (*Shen et al., 2015*) highlighted the issue of PKR and its capability of detecting it as per Table 8. The study spotted nine papers that scrutinized the PKR evasion using static analysis, and only one papers scrutinized PKR using dynamic analysis.

– IDR Identifier Renaming Evasion Detection:

DroidOlytics (*Faruki et al., 2013*), AndroSimilar (*Faruki et al., 2015d*), Droidkin (*Gonzalez, Stakhanova & Ghorbani, 2014*), Kuhnel (*Kuhnel, Smieschek & Meyer, 2015*), Triggerscope (*Fratantonio et al., 2016*), AAMO (*Preda & Maggi, 2016*), and Battista (*Battista et al., 2016*) claim they can detect IDR evasion by using their static Android malware detection frameworks as presented in Table 8. Nevertheless, many other researchers inadequately evaluate its robustness against IDR evasion. Table 8 demonstrates the issue of assuring the Android malware detection frameworks' robustness against IDR evasion and scrutinizes the researchers' framework against IDR evasion techniques.

In summary, most Android malware detection frameworks based on static analysis can detect package transformation techniques (RPK, PKR, and IDR). However, most detection frameworks based on dynamic and hybrid analysis inadequately evaluate or report their resilience against IDR evasion techniques. The study spotted 20 papers that scrutinized the RPK evasion using static analysis, and only 10 papers scrutinized IDR. The study spotted nine papers that scrutinized the IDR evasion using static analysis, and only one paper scrutinized IDR using dynamic analysis.

(b) Encryption Transformation Evasion Detection:

Static analysis detects encryption evasion techniques; many studies, such as DexHunter (*Zhang, Luo & Yin, 2015*), DroidKin (*Gonzalez, Stakhanova & Ghorbani, 2014*), Sherlockdroid (*Apvrille & Apvrille, 2015*), Kuhnel (*Kuhnel, Smieschek & Meyer, 2015*), and AAMO (*Preda & Maggi, 2016*), have proved that they detect the three encryption evasions (DEN, BEN, and PEN). Static based detection studies, such as AndroSimilar (*Faruki et al., 2015d*), MysteryChecker (*Jeong et al., 2014*), DroidKin (*Gonzalez, Stakhanova & Ghorbani, 2014*), SherlockDroid (*Apvrille & Apvrille, 2015*), Kuhnel (*Kuhnel, Smieschek & Meyer, 2015*), Shen (*Shen et al., 2015*), and AAMO (*Preda & Maggi, 2016*), are able to detect DEN evasions. Likewise, Soh (*Soh et al., 2015*) and Q-floid (*Castellanos et al., 2016*) claimed robustness against BEN evasion. The dynamic analysis based detection DwroidDump (*Kim, Kwak & Ryou, 2015*) used code extraction executable code from the memory of Dalvik Virtual Machine (DVM) instead of using a decompilation tool, which is subject to obstruction by the three encryption evasions techniques as shown in Table 8. Nevertheless, the RiskRanker (*Grace et al.,*

**Table 8 Polymorphism evaluation of frameworks.**

| | | | Android malware detection frameworks | | |
|---|---|---|---|---|---|
| | | | Static | Dynamic | Hybrid |
| Polymorphism | Package transformation | Repacking (RPK) | DroidMat (*Wu et al., 2012*), DroidOLytics (*Faruki et al., 2013*), ViewDroid (*Zhang et al., 2014*), DroidGraph (*Kwon et al., 2014*), MysteryChecker (*Jeong et al., 2014*), Chen (*Chen et al., 2015*), Dempster–Shafe (*Du, Wang & Wang, 2015*), DroidExec (*Wei et al., 2015*), AnDarwin and DNADroid (*Crussell, Gibler & Chen, 2015*), AndroSimilar (*Faruki et al., 2015a*), Ngrams (*Canfora et al., 2015a*), DroidEagle (*Sun, Li & Lui, 2015*), Droidkin (*Gonzalez, Stakhanova & Ghorbani, 2014*), AAMO (*Preda & Maggi, 2016*), AndroDet (*Mirzaei et al., 2019*), Karbab (*Karbab & Debbabi, 2021*), Amin (*Amin et al., 2020*), BLADE (*Sihag et al., 2021*), Dadidroid (*Ikram, Beaume & Kaafar, 2019*), Obfusifier (*Li et al., 2019*) | Soh (*Soh et al., 2015*) | NA |
| | | Package Renaming (PKR) | DroidMat (*Wu et al., 2012*), DroidOLytics (*Faruki et al., 2013*), Chen (*Chen et al., 2015*), AnDarwin and DNADroid (*Crussell, Gibler & Chen, 2015*), AndroSimilar (*Faruki et al., 2015d*), Ngrams (*Canfora et al., 2015a*), Droidkin (*Gonzalez, Stakhanova & Ghorbani, 2014*), Gurulian (*Gurulian et al., 2016*), AAMO (*Preda & Maggi, 2016*), Battista (*Battista et al., 2016*), Obsifier (*Li et al., 2019*), Kim (*Kim et al., 2019*), Dadidroid (*Ikram, Beaume & Kaafar, 2019*), Balde (*Sihag et al., 2021*), Dharmalingam (*Dharmalingam & Palanisamy, 2021*), Karbab (*Karbab & Debbabi, 2021*), AndrODet (*Mirzaei et al., 2019*) | Soh (*Soh et al., 2015*) | Abaid (*Abaid, Kaafar & Jha, 2017*) |
| | | Identifier Renaming (IDR) | DroidMat (*Wu et al., 2012*), Chen (*Chen et al., 2015*), Ngrams (*Canfora et al., 2015a*), SeqMalSpec -Sufatrio (*Sufatrio et al., 2015a*), Droidkin (*Gonzalez, Stakhanova & Ghorbani, 2014*), Shen (*Shen et al., 2015*), Kuhnel (*Kuhnel, Smieschek & Meyer, 2015*), Gurulian (*Gurulian et al., 2016*), AAMO (*Preda & Maggi, 2016*), Battista (*Battista et al., 2016*), AndrODet (*Mirzaei et al., 2019*), Karbab (*Karbab & Debbabi, 2021*), Dharmalingam (*Dharmalingam & Palanisamy, 2021*), Kim (*Kim et al., 2019*), Dadidroid (*Ikram, Beaume & Kaafar, 2019*), Obfusifier (*Li et al., 2019*) | Soh (*Soh et al., 2015*), Wu, 2015 (*Wu et al., 2015*) | |
| | Encryption | Data Encryption (DEN) | DroidMat (*Wu et al., 2012*), MysteryChecker (*Jeong et al., 2014*), Dexhunter (*Zhang, Luo & Yin, 2015*), AndroSimilar (*Faruki et al., 2015d*), Droidkin (*Gonzalez, Stakhanova & Ghorbani, 2014*), Shen (*Shen et al., 2015*), SherlockDroid (*Apvrille & Apvrille, 2015*), Kuhnel (*Kuhnel, Smieschek & Meyer, 2015*), AAMO (*Preda & Maggi, 2016*), AndrODet (*Mirzaei et al., 2019*), DroidSieve (*Suarez-Tangil et al., 2017*), AndrODet (*Mirzaei et al., 2019*), Karbab (*Karbab & Debbabi, 2021*), Intdroid (*Zou et al., 2021*), BLADE (*Sihag et al., 2021*), Alazab (*Alazab et al., 2020*), Kim (*Kim et al., 2019*), Dadidroid (*Ikram, Beaume & Kaafar, 2019*) | DwroidDump (*Kim, Kwak & Ryou, 2015*) | RiskRanker (*Grace et al., 2012*), Mobile-Sandbox (*Spreitzenbarth et al., 2015*) |
| | | Bytecode Encryption (BEN) | DroidMat (*Wu et al., 2012*), DroidOLytics (*Faruki et al., 2013*), Dexhunter (*Zhang, Luo & Yin, 2013*), DroidAPIMiner (*Aafer, Du & Yin, 2013*), MysteryChecker (*Jeong et al., 2014*), Dexhunter (*Zhang, Luo & Yin, 2015*), Droidkin (*Gonzalez, Stakhanova & Ghorbani, 2014*), SherlockDroid (*Apvrille & Apvrille, 2015*), Kuhnel (*Kuhnel, Smieschek & Meyer, 2015*), AAMO (*Preda & Maggi, 2016*), Wang (*Wang et al., 2016*), DroidSieve (*Suarez-Tangil et al., 2017*), Intdroid (*Zou et al., 2021*), Dharmalingam (*Dharmalingam & Palanisamy, 2021*), Dadidroid (*Ikram, Beaume & Kaafar, 2019*) | DwroidDump (*Kim, Kwak & Ryou, 2015*) | RiskRanker (*Grace et al., 2012*), Mobile-Sandbox (*Spreitzenbarth et al., 2015*) |
| | | Payload Encryption (PEN) | DroidMat (*Wu et al., 2012*), DroidOLytics (*Faruki et al., 2013*), Dexhunter (*Zhang, Luo & Yin, 2013*), Droidkin (*Gonzalez, Stakhanova & Ghorbani, 2014*), SherlockDroid (*Apvrille & Apvrille, 2015*), Kuhnel (*Kuhnel, Smieschek & Meyer, 2015*), DroidSieve (*Suarez-Tangil et al., 2017*), AAMO (*Preda & Maggi, 2016*), Intdroid (*Zou et al., 2021*), Karbab (*Karbab & Debbabi, 2021*), Dadidroid (*Ikram, Beaume & Kaafar, 2019*) | DwroidDump (*Kim, Kwak & Ryou, 2015*) | RiskRanker (*Grace et al., 2012*), Mobile-Sandbox (*Spreitzenbarth et al., 2015*) |

*2012*) hybrid based detection framework successfully detected DEN, BEN, and PEN. Hybrid detection frameworks such as RiskRanker (*Grace et al., 2012*), AMDetector (*Zhao et al., 2014*), MARVIN (*Lindorfer, Neugschwandtner & Platzer, 2015*), and Mobile-Sandbox (*Spreitzenbarth et al., 2015*) evaluated their frameworks against DEN evasion; they claim the ability to detect BEN evasion techniques. Two dynamic detections papers evaluate their frameworks against RPK evasion techniques: Soh (*Soh et al., 2015*) and Wu 2015 (*Wu et al., 2015*). Likewise, DwroidDump (*Kim, Kwak & Ryou, 2015*) examines its framework against encryption evasion techniques. *Kumawat, Sharma & Kumawat (2017)* also developed a system to detect cryptographic vulnerabilities in Android applications and to detect malware. This study spotted seven papers that scrutinized the DEN evasion using static analysis, only one paper scrutinized DEN using dynamic analysis, and two papers scrutinized DEN using hybrid analysis. However, this study spotted six papers that scrutinized the BEN evasion using static analysis, only one paper scrutinized BEN using dynamic analysis, and two papers scrutinized BEN using hybrid analysis. In addition, this study spotted five papers that scrutinized the PEN evasion using static analysis, only one paper scrutinized PEN using dynamic analysis, and two papers scrutinized PEN using hybrid analysis.

## Metamorphism evasion detection

Table 8 represents static, dynamic, and hybrid-based Android malware detection frameworks and their robustness against metamorphism evasion detection techniques.

(a) Code Obfuscation Detection:

Code obfuscation consists of CRE, CIN, and DCI; we explain each evasion detection framework in the following list:

– CRE - Code Reordering Evasion Detection:

ResDroid (*Shao et al., 2014*), AnDarwin (*Crussell, Gibler & Chen, 2015*), and Seqmalspec (*Sufatrio et al., 2015a*) proposed static analysis based detection and managed to detect CRE evasion. Likewise, Q-floid (*Castellanos et al., 2016*) detected CRE using the dynamic sandboxing methodology. Mobile-Sandbox (*Spreitzenbarth et al., 2015*) hybrid based detection frameworks detect CRE evasions. Nonetheless, CRE evades ngrams (*Canfora et al., 2015a*) and Elish (*Elish et al., 2015*) static detection frameworks, which results in many false negatives (FN), as shown in Table 9. This study spotted 17 papers that scrutinized the CRE evasion using static analysis, only two papers scrutinized CRE using dynamic analysis, and four papers scrutinized CRE using hybrid analysis.

– CIN - Call Indirections Evasion Detection:

As shown in Table 9, the CIN evasion technique successfully evades the call graph based Android malware detection frameworks (*Chenxiong et al., 2015*; *Poeplau et al., 2014*; *Wu et al., 2016*). Despite the fact that many static frameworks easily detect CIN evasion

**Table 9 Metamorphism evaluation of frameworks.**

| | | Android malware detection frameworks | | |
|---|---|---|---|---|
| | | **Static** | **Dynamic** | **Hybrid** |
| Metamorphism | Code obfuscation | Code Reordering (CRE) | DroidOLytics (*Faruki et al., 2013*), DroidGraph (*Kwon et al., 2014*), MysteryChecker (*Jeong et al., 2014*), ResDroid (*Shao et al., 2014*), Apk Auditor (*Talha, Alper & Aydin, 2015*), Dempster–Shafe (*Du, Wang & Wang, 2015*), Dexhunter (*Zhang, Luo & Yin, 2015*), DroidExec (*Wei et al., 2015*), AnDarwin and DNADroid (*Crussell, Gibler & Chen, 2015*), AndroSimilar (*Faruki et al., 2015d*), SeqMalSpec -Sufatrio et al., 2015a), DroidEagle (*Sun, Li & Lui, 2015*), Shen (*Shen et al., 2015*), Gurulian (*Gurulian et al., 2016*), AAMO (*Preda & Maggi, 2016*), Wang (*Wang et al., 2016*), MocDroid (*Martín, Menéndez & Camacho, 2016*), Battista (*Battista et al., 2016*), DroidSieve (*Suarez-Tangil et al., 2017*), AndrODet (*Mirzaei et al., 2019*), Karbab (*Karbab & Debbabi, 2021*), Obfusifier (*Li et al., 2019*), Kim (*Kim et al., 2019*), Dadidroid (*Ikram, Beaume & Kâafar, 2019*) | Soh (*Soh et al., 2015*), Q-floid (*Castellanos et al., 2016*) | RiskRanker (*Grace et al., 2012*), MDetector (*Zhao et al., 2014*), MARVIN (*Lindorfer, Neugschwandtner & Platzer, 2015*), Mobile-Sandbox (*Spreitzenbarth et al., 2015*) |
| | | Call Indirections (CIN) | DroidOLytics (*Faruki et al., 2013*), DroidGraph (*Kwon et al., 2014*), AdDetect (*Narayanan, Chen & Chan, 2014*), Apk Auditor (*Talha, Alper & Aydin, 2015*), Dempster–Shafe (*Du, Wang & Wang, 2015*), Dexhunter (*Zhang, Luo & Yin, 2015*), DroidExec (*Wei et al., 2015*), AnDarwin and DNADroid (*Crussell, Gibler & Chen, 2015*), AndroSimilar (*Faruki et al., 2015d*), DroidEagle (*Sun, Li & Lui, 2015*), Shen (*Shen et al., 2015*), Gurulian (*Gurulian et al., 2016*), AAMO (*Preda & Maggi, 2016*), Wang (*Wang et al., 2016*), MocDroid (*Martín, Menéndez & Camacho, 2016*), Battista (*Battista et al., 2016*), DroidSieve (*Suarez-Tangil et al., 2017*), AndrODet (*Mirzaei et al., 2019*), Karbab (*Karbab & Debbabi, 2021*), Obfusifier (*Li et al., 2019*) | Soh (*Soh et al., 2015*), Q-floid (*Castellanos et al., 2016*) | RiskRanker (*Grace et al., 2012*), MDetector (*Zhao et al., 2014*), MARVIN (*Lindorfer, Neugschwandtner & Platzer, 2015*) |
| | | Dead Code Insertion (DCI) | DroidOLytics (*Faruki et al., 2013*), DroidGraph (*Kwon et al., 2014*), AdDetect (*Narayanan, Chen & Chan, 2014*), Apk Auditor (*Talha, Alper & Aydin, 2015*), Dempster–Shafe (*Du, Wang & Wang, 2015*), Dexhunter (*Zhang, Luo & Yin, 2015*), DroidExec (*Wei et al., 2015*), AndroSimilar (*Faruki et al., 2015d*), DroidEagle (*Sun, Li & Lui, 2015*), Shen (*Shen et al., 2015*), Gurulian (*Gurulian et al., 2016*), AAMO (*Preda & Maggi, 2016*), Wang (*Wang et al., 2016*), MocDroid (*Martín, Menéndez & Camacho, 2016*), Battista (*Battista et al., 2016*), DroidSieve (*Suarez-Tangil et al., 2017*), AndrODet (*Mirzaei et al., 2019*), Karbab (*Karbab & Debbabi, 2021*), Obfusifier (*Li et al., 2019*), Alazab (*Alazab et al., 2020*), Pektas (*Pektaş & Acarman, 2020*) | No dynamic frameworks | RiskRanker (*Grace et al., 2012*), ARIGUMA (*Zhong et al., 2013*), AMDetector (*Zhao et al., 2014*), MARVIN (*Lindorfer, Neugschwandtner & Platzer, 2015*) |

| | | Android malware detection frameworks | | |
| --- | --- | --- | --- | --- |
| | | **Static** | **Dynamic** | **Hybrid** |
| Advanced Code transformation | Native Exploits (NEX) | DroidAPIMiner (*Aafer, Du & Yin, 2013*), AdDetect (*Narayanan, Chen & Chan, 2014*) | DroidBarrier (*Almohri, Yao & Kafura, 2014*) | MARVIN (*Lindorfer, Neugschwandtner & Platzer, 2015*) |
| | Function Inlining and Outlining (FIO): | AAMO (*Preda & Maggi, 2016*) | No Dynamic frameworks | No hybrid frameworks |
| | Reflection API (REF) | Juxtapp (*Hanna et al., 2013*), DroidAPIMiner (*Aafer, Du & Yin, 2013*), Dexhunter (*Zhang, Luo & Yin, 2015*), SherlockDroid (*Apvrille & Apvrille, 2015*), Kuhnel (*Kuhnel, Smieschek & Meyer, 2015*), AAMO (*Preda & Maggi, 2016*), DroidSieve (*Suarez-Tangil et al., 2017*), Yang (*Yang et al., 2021*), BLADE (*Sihag et al., 2021*), Karbab (*Karbab & Debbabi, 2021*) | Maier (*Maier, Protsenko & Müller, 2015*), EnDroid (*Feng et al., 2018*) | RiskRanker (*Grace et al., 2012*), StaDyna (*Zhauniarovich et al., 2015*) |
| | Dynamic code loading (DCL) | DroidAPIMiner (*Aafer, Du & Yin, 2013*), Yerima (*Yerima, Sezer & Muttik, 2014*), ResDroid (*Shao et al., 2014*), Poeplau (*Poeplau et al., 2014*), Dexhunter (*Zhang, Luo & Yin, 2015*), Grab 'n Run Falsina (*Falsina et al., 2015*), DroidSieve (*Suarez-Tangil et al., 2017*), Yang (*Yang et al., 2021*) | Maier (*Maier, Protsenko & Müller, 2015*), EnDroid (*Feng et al., 2018*) | RiskRanker (*Grace et al., 2012*), StaDyna (*Zhauniarovich et al., 2015*), Abaid (*Abaid, Kaafar & Jha, 2017*) |
| | Anti-debugging (ADE) | Dexhunter (*Zhang, Luo & Yin, 2015*) | | MARVIN (*Lindorfer, Neugschwandtner & Platzer, 2015*) |
| Anti-emulator | Virtual Machine Aware (VMA) | No static frameworks | Tao (*Tao et al, 2012*), DroidScope (*Yan & Yin, 2012*), Pektas (*Pektas & Acarman, 2014*), Maier (*Maier, Protsenko & Müller, 2015*), Singh (*Singh, Mishra & Singh, 2015*), GroddDroid (*Abraham et al., 2015*), Alzaylaee (*Alzaylaee, Yerima & Sezer, 2017*) | RiskRanker (*Grace et al., 2012*), Petsas (*Petsas et al., 2014*), Tap-Wave-Rub (*Shrestha et al., 2015*) |
| | Programmed Interaction Detection (PID) | No static frameworks | Chaugule (*Chaugule, Xu & Zhu, 2011*), Singh (*Singh, Mishra & Singh, 2015*), GroddDroid (*Abraham et al., 2015*), Diao (*Diao et al., 2016*) | Tap-Wave-Rub (*Shrestha et al., 2015*) |

(*Faruki et al., 2015d*; *Faruki et al., 2013*; *Gurulian et al., 2016*; *Kwon et al., 2014*; *Martín, Menéndez & Camacho, 2016*; *Narayanan, Chen & Chan, 2014*; *Wei et al., 2015*; *Zhang, Luo & Yin, 2015*), CIN still defeats other frameworks such as APK Auditor (*Talha, Alper & Aydin, 2015*), Andro-Tracer (*Kang et al., 2015*), ngrams (*Canfora et al., 2015a*), Elsih (*Elish et al., 2015*) and Wu (*Wu et al., 2016*). Few dynamic analysis based detection frameworks (*Castellanos et al., 2016*; *Soh et al., 2015*) and hybrid detection frameworks such as (*Grace et al., 2012*; *Lindorfer, Neugschwandtner & Platzer, 2015*; *Zhao et al., 2014*) can detect-Call Indirections Evasion CIN. *Choliy, Li & Gao (2017)* developed a system called ACTS (App topologiCal signature through graphleT Sampling) in which they detected obfuscated function calls in malware samples. This study spotted 15 papers that scrutinized the CIN evasion using static analysis, only two papers scrutinized CIN using dynamic analysis, and three papers scrutinized CIN using hybrid analysis.

– DCI - Dead Code Insertion Evasion Detection:

AnDarwin (*Crussell, Gibler & Chen, 2015*) conducted dead code insertion detection experiments based on code similarity. AnDarwin reported that it is less robust to dead code insertion transformation (*Crussell, Gibler & Chen, 2015*) that adopts code's similarity approach with semantic analysis, as shown in Table 9. The similarity approach examines the distance vector values using semantic analysis. The distance vector increases with the code alteration between the original and after dead code insertion obfuscation. This study spotted 14 papers that scrutinized the DCI evasion using static analysis, and four papers scrutinized DCI using hybrid analysis.

In general, the dynamic analysis framework Q-floid (*Castellanos et al., 2016*) introduces the Qualitative Data Flow Graph (QDFG) to analyze the dynamic behaviour of a suspicious app. It states that it detects code obfuscation, basing this assumption on PC-based malware detection using Q-floid (*Castellanos et al., 2016*). It detects code obfuscation transformation using the QDFG (*Banescu et al., 2015*; *Wüchner, Ochoa & Pretschner, 2015*). However, it claims that Q-floid (*Castellanos et al., 2016*) inadequately detects Android malware when restricting using monitoring services. MysteryChecker (*Jeong et al., 2014*) proposes a novel software-based attestation approach to detect the repackaged malware with code obfuscation and a randomly selected encryption chain. Likewise, Gurulian (*Gurulian et al., 2016*) introduces a DCI evasion resilient framework by maintaining the attack vector; similarly, DroidOLytics (*Faruki et al., 2013*) uses statistical similarity to detect application repackaging and code obfuscation. It builds a signature repository that changes its length dynamically for code cloning detection. AndroSimilar (*Faruki et al., 2015d*) uses signature-based detection and attains 76% accuracy, but its detection rate of repacking and code obfuscation transformation evasions is relatively low. Until today, AndRODet (*Mirzaei et al., 2019*) adopts static analysis to detect Android malware applications with CRE, CIN, and DCI evasions; however, the average achieved performance for detection CRE, CIN, and DCI evasions is 63%.

(b) Advanced Code Transformation Detection:

It consists of NEX, FIO, REF, DCL, and ADE evasions explained in this section.

– NEX Evasion Detection:

DroidAPIMiner (*Aafer, Du & Yin, 2013*) uses static analysis to detect NEX evasion and, as listed in Table 9, claims success; likewise, the dynamic analysis DroidBarrier (*Almohri, Yao & Kafura, 2014*) and hybrid analysis MARVIN (*Lindorfer, Neugschwandtner & Platzer, 2015*) claim the same. In contrast, many static frameworks such as AdDetect (*Narayanan, Chen & Chan, 2014*), APK Auditor (*Talha, Alper & Aydin, 2015*), Andro-Tracer (*Kang et al., 2015*), and ngrams (*Canfora et al., 2015a*) stated their limitations in countermeasures of NEX evasion as shown in Table 9. This study spotted one paper that scrutinized the CIN evasion using static analysis, one paper scrutinized CIN using dynamic analysis, and one paper scrutinized CIN using hybrid analysis.

– FIO Evasion Detection:

AAMO (*Preda & Maggi, 2016*) evaluates anti-virus packages *vs* function inlining and outlining FIO evasion, as shown in Table 9. However, dynamic analysis and hybrid analyses inadequately consider the evaluation of their framework against FIO evasion. This study spotted one paper that scrutinized the FIO evasion using static analysis, and two papers scrutinized FIO using dynamic analysis.

– REF Evasion Detection:

As shown in Table 9, many static analysis frameworks examine the robustness of their detection frameworks against REF evasion, such as DroidAPIMiner (*Aafer, Du & Yin, 2013*), DexHunter (*Zhang, Luo & Yin, 2015*), SherLockDroid (*Apvrille & Apvrille, 2015*), Kuhnel (*Kuhnel, Smieschek & Meyer, 2015*), DroidRA (*Li et al., 2016*), and AAMO. Likewise, Maier (*Maier, Protsenko & Müller, 2015*), which uses Dynamic analysis, RiskRanker (*Grace et al., 2012*), and StaDyna (*Zhauniarovich et al., 2015*), which use hybrid analysis, study REF evasion detection using dynamic and hybrid analysis based detection techniques. This study spotted six papers that scrutinized the REF evasion using static analysis, only two papers scrutinized REF using dynamic analysis, and two papers scrutinized REF using hybrid analysis.

– DCL Evasion Detection:

Some Android malware detection frameworks propose and evaluate their methods to detect DCL evasion, for instance, DroidAPIMiner (*Aafer, Du & Yin, 2013*), Poeplau (*Poeplau et al., 2014*), Dexhunter, Maier (*Maier, Protsenko & Müller, 2015*), RiskRanker (*Grace et al., 2012*), and StaDyna (*Zhauniarovich et al., 2015*). However, AndroSimilar (*Faruki et al., 2015d*) insufficiently evaluates its mechanism against dynamic code loading, reflection, and other transformation techniques, as shown in Table 9. This study spotted four papers that scrutinized the DCL evasion using static analysis, only

two papers scrutinized DCL using dynamic analysis, and two papers scrutinized DCL using hybrid analysis.

ADE Evasion Detection: Only the static analysis DexHunter (*Zhang, Luo & Yin, 2015*) considered the ADE evasion technique in evaluating the framework. On the contrary, the dynamic analysis Q-floid (*Castellanos et al., 2016*) reported ineffective ADE evasion detection, as shown in Table 9. This study spotted one paper that scrutinized the ADE evasion using static analysis.

– Anti-emulation Detection

Anti-emulation evasions consist of VMA and PID evasion techniques; the following is the insight of detection framework analysis:

– VMA Evasion Detection:

As a countermeasure for the VMA evasion technique, researchers (*David & Netanyahu, 2015*; *Mutti et al., 2015*) equip an emulator sandbox with physical devices to dynamically run the application analyzes. *Dietzel (2014)*, *Gajrani et al. (2015)*, and *Hu & Xiao (2014)* propose a fake response agent, which feeds the in the dynamic analysis based testing and a masquerade emulator as a physical device. In late 2015 and the beginning of 2016, several studies analyze the nature of anti-emulation malware with false values about the environment request. This study spotted six papers that scrutinized the WMA using dynamic analysis, and three papers scrutinized WMA using hybrid analysis.

Singh (*Singh, Mishra & Singh, 2015*) enhances the dynamic malware detection robustness, using anti-emulator and user interaction detection. Petsas (*Petsas et al., 2014*) proposes countermeasures for different evasion detections, such as anti-emulation using realistic sensor simulation and IMEI modification. However, it inadequately evaluates this countermeasure. Dynalog (*Alzaylaee, Yerima & Sezer, 2016*) proposes a performance-enhanced Android malware dynamic analysis that uses the emulation tool, subject to emulation detection evasions. Likewise, Dynalog (*Alzaylaee, Yerima & Sezer, 2016*) highlights the issue of dynamic analysis evasion without proposing a solution. To overcome VMA evasion, Vidas (*Vidas et al., 2014*) proposes system logs and network traffic classification features using a physical device A5 instead of emulator evasion techniques. Some studies only hoist the red flag to indicate that neither enough malware samples nor test benches exist for examining anti-emulation evasion (works such as *Chaugule, Xu & Zhu (2011)* and *Tao et al. (2012)*). Nevertheless, *Maier, Protsenko & Müller (2015)* studied VWA evasion and proposed a solution based on comparing the behaviour of the APK when installing on a physical device and emulator, as shown in Table 9.

– PID Evasion Detection:

Programmed Interaction Detection is fortunate to evade automated dynamic analysis using the inherent difference between key runner and human interaction patterns

(*Diao et al., 2016*). Instead of relying on identifying old virtualization or emulation techniques, *Diao et al. (2016)* focuses on detecting the automated gesture, which simulates user input, to conclude whether the application is under analysis or working under normal conditions, as shown in Table 9. As this anti-emulation evasion targeted sandboxing, which takes place during the dynamic analysis based detection, most of the efforts to countermeasure this type of evasion have used dynamic or hybrid analysis detection frameworks. This study spotted four papers that scrutinized the PID using dynamic analysis, and one paper scrutinized PID using hybrid analysis.

# DISCUSSION

In this section, this paper synthesizes the last decade's Android malware detection framework using three methodologies. First is identifying the evasions techniques requiring more attention from the research community. The second represents the potential evasion resilient detection techniques by reporting each framework's number of considered evasion techniques. The third summarizes the three types of Android application analysis with the number of frameworks that evaluated evasions techniques by bubble plot chart. Finally, we provide a to-do list and learned lessons from all the examined frameworks.

The static analysis radar graph shown in Fig. 5 signifies the evasion detection capabilities of static based detection. It serves to understand the evaluation of the static analysis based detection frameworks.

Figure 5 presents the static analysis based Android malware detection frameworks using the radar graph approach. The radar graph represents the number of frameworks in circular layers, starting with the outside circle, which means zero frameworks. The second circular layer represents five frameworks. The inner-circle layer represents the largest number of frameworks that examined evasion techniques. Each evasion technique is labelled point such as PID, WMA, ADE, DCL, *etc*. Besides each point number representing the number of Android malware detection frameworks that evaluated its proposed model against this evasion technique or point in the radar graph. For example, 15 malware detection frameworks consider the RPR evasion technique; thus, the RPK label points to 15, as displayed in Fig. 5. The evasion techniques that avoid Android malware detection using VMA and PID have zero values besides their points, as shown in Fig. 5.

We selected the Radar graph to demonstrate that static detection studies could detect package transformation evasions and basic code obfuscation; however, advanced transformation techniques and anti-emulation were neither studied nor evaluated. Concerning DCL, Pektas (*Pektas & Acarman, 2014*), in 2014, detected anti-emulation evasion by using a dynamic analyzing tool developed just to deal with the DCL evasion malware samples, which achieved 92% accuracy. Many researchers avoid using dynamic-based detection techniques because they are time-consuming and risk installing malware into their testing devices. In Mobile-Sandbox (*Spreitzenbarth et al., 2015*), the dynamic analysis required an average of 18 min to accomplish the dynamic analysis tasks. This time depends on the size of the APK file and the dynamic analysis server hardware specifications.

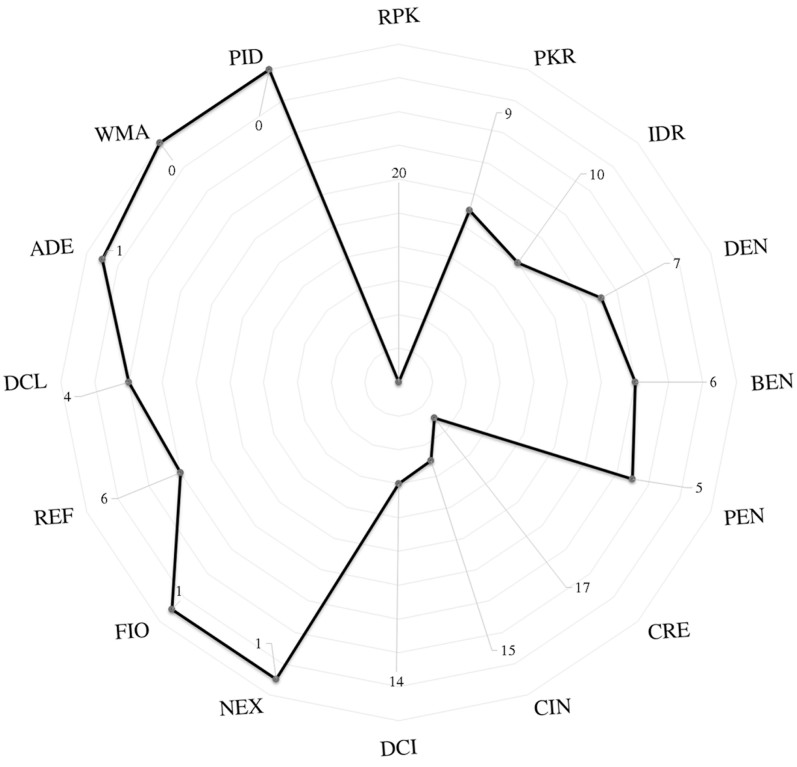

**Figure 5  Evasion techniques radar for static frameworks.**

Until today, many static analysis researchers depends on permissions (*Arora, Peddoju & Conti, 2019*; *Dharmalingam & Palanisamy, 2021*; *Li et al., 2018*; *Şahin et al., 2021*); however, many are relying on API calls (*Alazab et al., 2020*; *Jung et al., 2018*; *Maiorca et al., 2017*; *Mirzaei et al., 2019*; *Pektaş & Acarman, 2020*; *Tiwari & Shukla, 2018*; *Zhang et al., 2020*; *Zhang, Breitinger & Baggili, 2016*; *Zou et al., 2021*) and deep code analysis and other types of features as discussed earlier in Android evasion detection frameworks section. Many of examined researches ignored the evasion techniques evaluation. Other frameworks assumed the impossibility of the evasion detection using static analysis and advise the research community to use dynamic analysis to detect it. Android Malware detection frameworks assumed their capability of detecting obfuscation techniques without evaluating their framework against obfuscated malware datasets. This paper examined 74 static frameworks, but only 35 research papers consider or evaluate their framework using at least one evasion technique, as shown in Fig. 6. The dynamic analysis evasion radar graph demonstrates the capabilities of dynamic analysis based.

Researchers assume that dynamic analysis covers all the simple obfuscations and transformation techniques. Hence many of the dynamic analysis frameworks (*Abuthawabeh & Mahmoud, 2019*; *Chen et al., 2018*; *de la Puerta et al., 2019*; *De Lorenzo et al., 2020*; *Feng et al., 2020*; *Feng et al., 2018*; *Pang et al., 2017*; *Sihag et al., 2021*; *Wang et al., 2019*) ignored the metamorphic evasion techniques. The overall performance

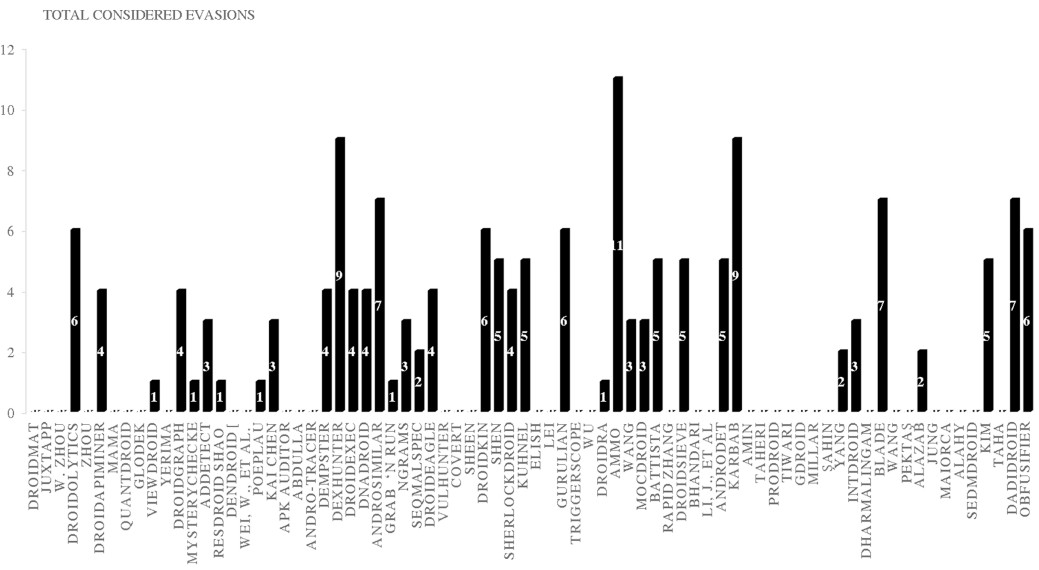

**Figure 6 Static analysis based frameworks and considered evasion.**

accuracy of the most current malware detection frameworks is measured against randomly selected malware samples representing certain malware families. If the randomly chosen malware families overlook evasion techniques, the selected malware insufficiently reflect the actual robustness of the proposed detection framework against evasion techniques; this was the main reason behind excluding the accuracy in evaluation tables. This paper examined 35 Android malware detection using different dynamic techniques. However, only 14 of 35 dynamic analysis based detection framworks have tried to include obfuscation into their evaluation processes, as shown in Fig. 7. Figure 8 shows the number of considered evasion techniques in each research is between 1 and 5 evasions. In its evaluation, *Soh et al. (2015)* considered three types of repackaging evasion, indirectly considered code reordering, and called indirection evasion. It defines many limitations to its approach and planned to consider the hybrid analysis in its future plan.

However, a few researchers evaluate their frameworks against specific evasion techniques, as reflected in the radar graph of the hybrid malware detection frameworks, as shown in Figs. 9 and 10. For instance, four frameworks claimed that their method detected the CRE and DCI evasions (*Grace et al., 2012*; *Lindorfer, Neugschwandtner & Platzer, 2015*; *Spreitzenbarth et al., 2015*; *Zhao et al., 2014*), and three frameworks claimed the detection of CIN (*Grace et al., 2012*; *Spreitzenbarth et al., 2015*; *Zhao et al., 2014*) and WMA (*Grace et al., 2012*; *Petsas et al., 2014*; *Yuan, Lu & Xue, 2016*). The hybrid based detection requires enormous effort to collect both static and dynamic characteristics and logs. RiskRanker (*Grace et al., 2012*) started highlighting the evasion problems and their impacts on detection accuracy. However, Petsas (*Petsas et al., 2014*) in 2014 and Tap-Wave-Rub (*Shrestha et al., 2015*) battled anti-emulation evasions and used the device

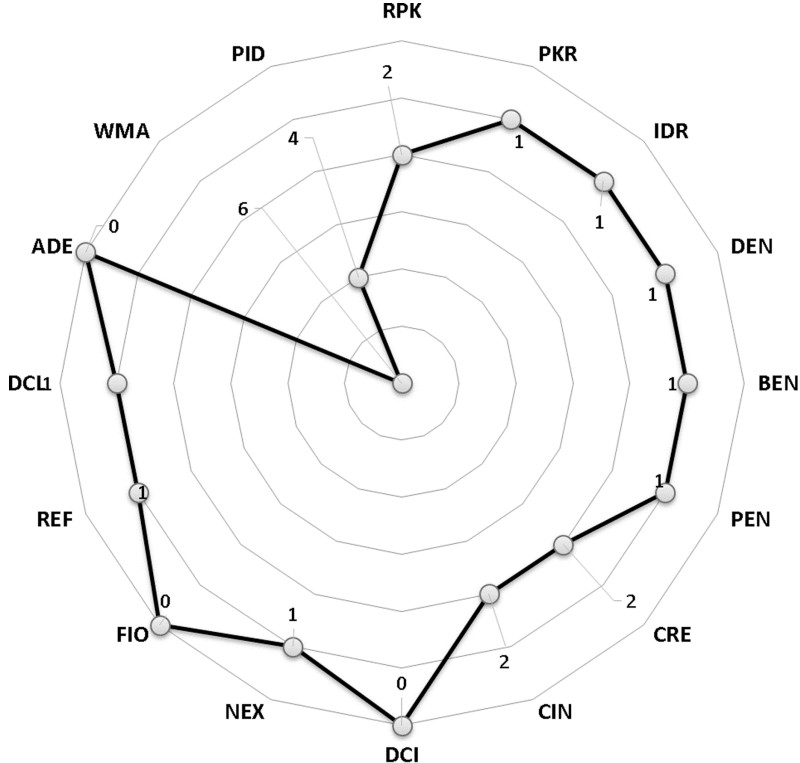

**─○─Number of Dynamic Frameworks vs Evasions Techniques**

**Figure 7  Dynamic analysis and evasion radar graph.**

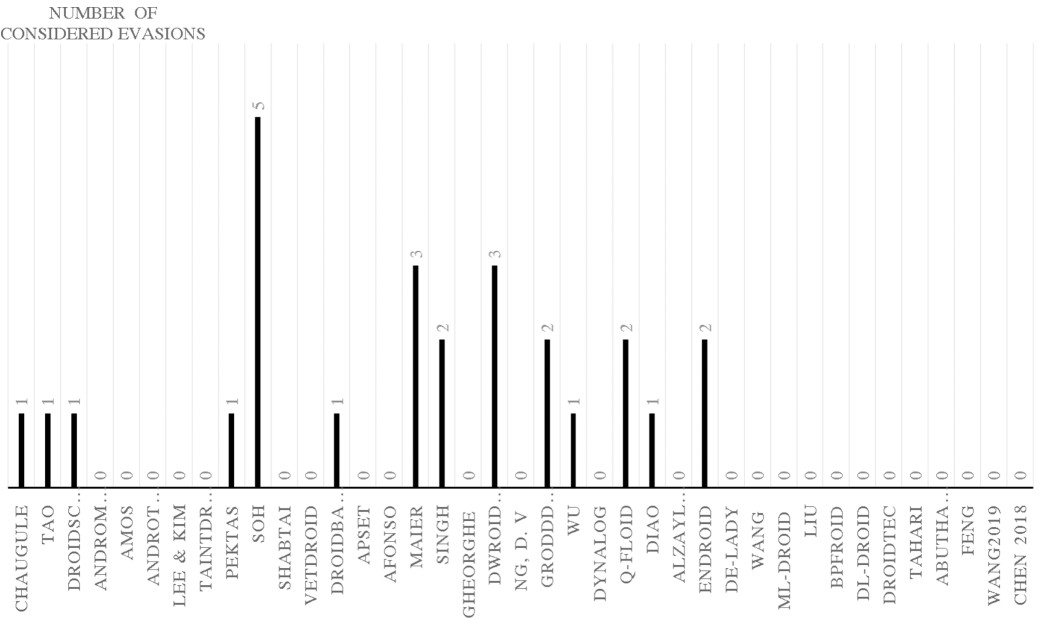

**Figure 8  Dynamic analysis based frameworks and considered evasion.**

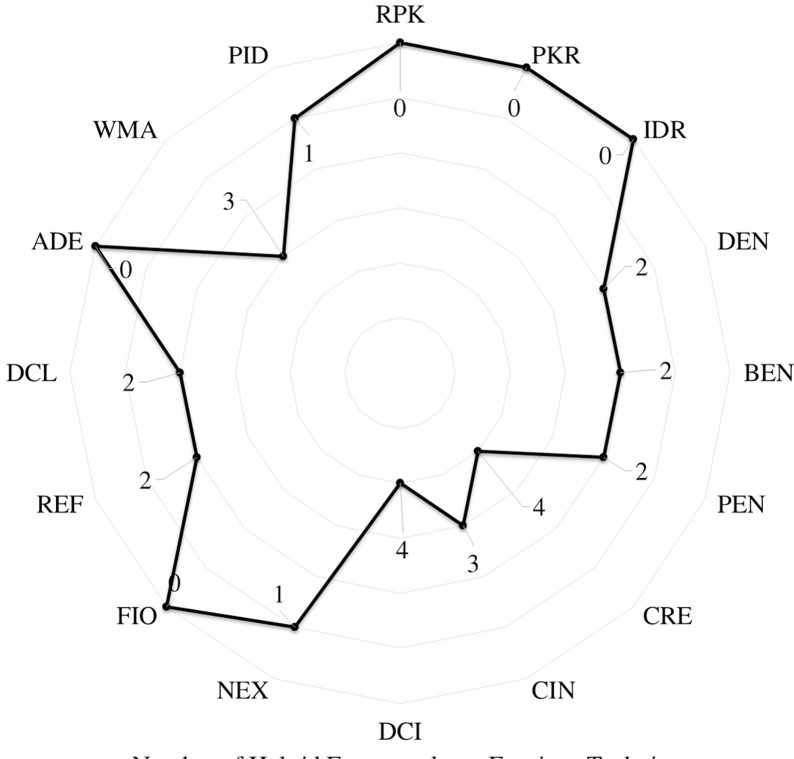

→ Number of Hybrid Frameworks vs Evasions Techniques

**Figure 9  Hybrid analysis and evasion radar.**

hardware (proximity sensor) to differentiate between maliciously driven actions and end-user physical interactions.

Most of the recent dynamic analysis researches (*Feng et al., 2020*; *Mahindru & Sangal, 2021*; *Sihag et al., 2021*) confirmed the ability to detect obfuscated Android malware. Unfortunately, none of dynamic analysis based detection has evaluated their framework using specific evasion techniques; most of dynamic analysis studies just randomly select from the publicly available Android malware datasets. For example, Droidetec (*Ma et al., 2020*) proposed a dynamic analysis based framework by analyzing the process behavior in an ordered manner. Still, the evaluation process was generic and included few malware families that exclude obfuscated malware.

The Hybrid analysis techniques are suggested by many researchers and have been set in their future plan to overcome the resiliency issue of complex obfuscation techniques. However, it is a shocking fact that the examined 26 Android malware detection frameworks using hybrid analysis, that only nine frameworks just consider few evasion techniques such as RiskRanker (*Grace et al., 2012*) that has initiated the issue in 2012, Mobile-Sandbox (*Hoffmann et al., 2016*), Marvin (*Lindorfer, Neugschwandtner & Platzer, 2015*). Recently some hybrid analysis based detection Puerta (*de la Puerta et al., 2019*), Surendrean (*Surendran, Thomas & Emmanuel, 2020*), Lu (*Lu et al., 2020*), Dhalaria (*Dhalaria & Gandotra, 2021*), Zhu (*Zhu et al., 2021*), Nawaz (*Nawaz, 2021*), Liu (*Liu et al., 2021*), PNSDroid (*Kandukuru & Sharma, 2018*), Bacci (*Bacci et al., 2018*), DAMBA

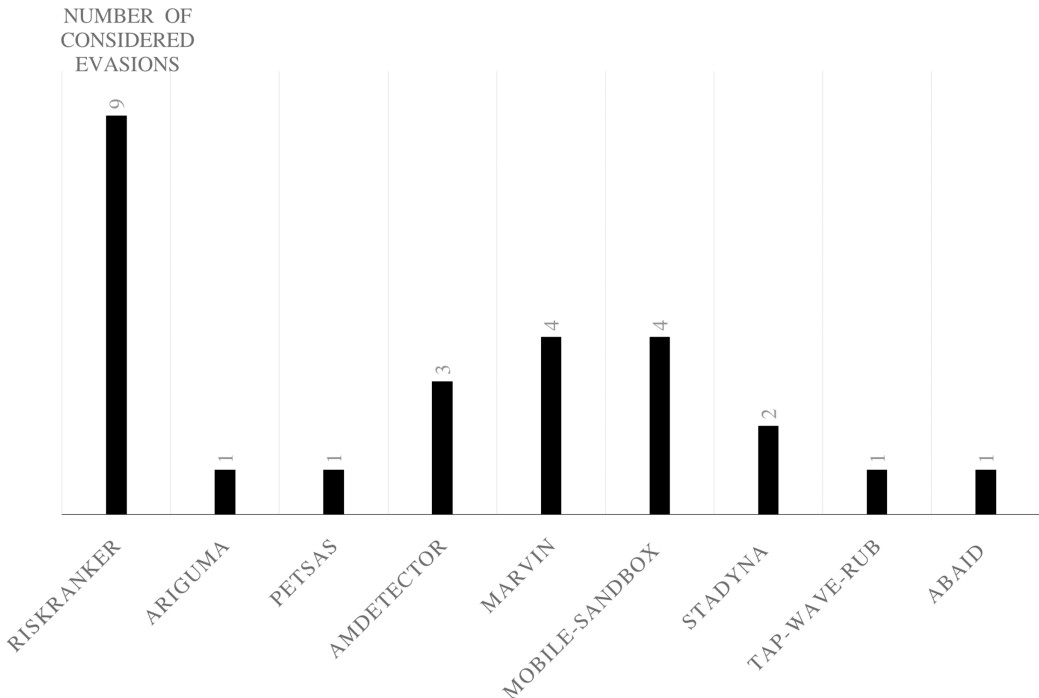

**Figure 10 Hybrid analysis based frameworks and considered evasion.**

(*Zhang et al., 2020*) has highlighted the complex evasions detection resiliency issue in their research literature; however, the proposed malware detection methods and experiments of excluded the obfuscated malware from their evaluation sheets.

The systematic evasion detection map is illustrated in Fig. 11; the horizontal axis represents each type of evasion in this study. The bubble size represents the accumulative number of detection techniques developed by the research community to fight each evasion technique. It is divided into three main categories in the vertical axis: static, dynamic, and hybrid detection techniques. For instance, the circle with the number "17" represents static Android malware detection frameworks, which consider CRE evasions on the framework evaluation process. As per the systematic map, the NEX, FIO, and ADE need more attention from the research community. Likewise, the overall dynamic analysis studies that considered evasion evaluation is shallow.

Researchers have concentrated on Android malware static analysis in the last few years, which requires less time and effort than dynamic analysis. They tried to overcome the static analysis weaknesses against evasion attacks, which is why many researchers evaluated their frameworks to check the anti-obfuscation capabilities, as presented in Fig. 11. Dynamic analysis researchers concentrated on avoiding virtualization detection and random interaction, which is the main reason for False Negative malware detection. Figure 11 shows the number of existing Android malware detection frameworks in each circle, which consider each evasion technique in the framework evaluations. It shows the necessity of more insights regarding evaluation against all types of evasions, as currently,

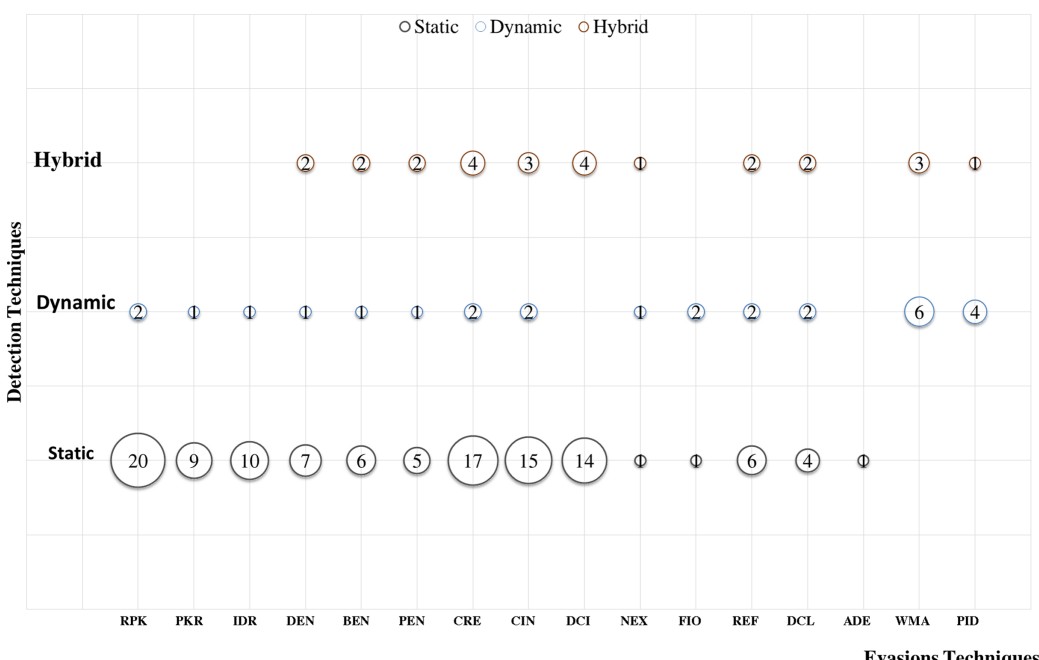

**Figure 11** **Systematic map of accumulative number of detection frameworks *vs* evasions techniques.**

available standard malware datasets cover some evasion techniques that are randomly
selected during evaluation. In summary, all the above investigations demonstrate the
absolute need for standard evasion benchmarking tools to evaluate the newly developed
frameworks against all evasion techniques.

# LESSONS LEARNED AND FUTURE DIRECTIONS

Android malware development is always one step ahead of malware detection techniques,
which means malware detection still requires many efforts to catch up with malware
development. To achieve this objective, we share several insights drawn from our analysis.

(a) Obfuscation dataset

One of the most important is to keep on updating and standardizing obfuscated
malware datasets. We recommend standardizing this dataset by the research community
trusted institutions and being available upon validated requests for research purposes.
Despite some available obfuscated datasets such as PRAGuard (*Karbab & Debbabi, 2021*)
sharing ten thousands obfuscated malware by obfuscating MalGnome and the Contagio
MiniDump dataset, however the PRAGuard stopped sharing the dataset starting from
April 2021.

(b) Obfuscation detection framework performance

The performance of the Android malware framework degraded over time since new
malware variants, and obfuscations techniques were generated PetaDroid (*Karbab &*

*Debbabi, 2021*). Hence, we recommend researchers extend their research to keep an eye on their framework performance over time.

(c) Metamorphism evasion:

Static detection is unable to detect most of the metamorphism evasion techniques because of the dynamic characteristics of metamorphism. However, there is still a lack of dynamic and hybrid frameworks to detect metamorphism evasions. It is therefore beneficial to focus more on developing dynamic and hybrid methods.

(d) Standard Evasion Benchmarking:

We suggest building a comprehensive and collaborative benchmarking framework for Android malware detection evasion techniques that aims to improve the quality of research and add to the body of knowledge in Android malware detection studies. The benchmark consists of a list of evasion techniques based on the detection methods that have been evaluated. As a result, detection methods are tested against a standardized list of malware evasion techniques to determine whether they are capable of detecting malware evasions.

(e) Android Exploits:

As mentioned earlier, Android is based on Linux OS; it has inherited Linux exploits. Recently, malware authors developed and published the Android exploit code Dirty-Cow CVE-2016-5195 (*Oester, 2016*). The Dirty Cow exploit has been existing in Linux since 2007; it affects all Android versions. Existing fixes for Linux exploits are inefficient; Android fixes are still expected from vendors like Google or Samsung. Researchers must study such exploits and recommend proper ways to fix newly discovered exploits. Additionally, researchers need to examine the Android operating system and identify potential exploits and offer solutions before attackers abuse such exploits.

(f) Code Integrity Verification:

Verification means that the application integrity is authenticated against repackaging by guaranteed third-party authentication authorities. *Vidas & Christin (2013)* proposed a simple mechanism that alleviates the specific problem of verifying the authenticity of an App to protect the user from repackaged apps that contain malicious code. Their approach is based on creating a simple public-key infrastructure backed by the domain name system. This area of research needs more attention compared to others. App integrity significantly increases the effort required for a successful attack. Under this new model, the attacker must either obtain the original publisher's secret signing key, control the publisher's web server, or commit a man-in-the-middle (MitM) attack on the publisher's DNS records and web server. The attacker must now conduct two successful attacks in all cases, and the secondary attack requires more effort than application repackaging. It is worth noting that code verification, and not code analysis, is recommended, as it is necessary to consider the

complexity of the available applications. Code verification does not require much effort, as it involves checking the code's integrity by using the public-key infrastructure.

(g) Process Authentication:

Some researchers leverage the process of model authentication to eliminate the need for an external Certification Authority (CA) that protects the system from many exploits (*Almohri, Yao & Kafura, 2014*). However, they are still unable to detect the payload that is downloaded to install malicious applications. For example, DroidBarrier is designed to prevent such installations by detecting their unauthenticated processes, thereby foiling this form of attack. However, DroidBarrier (*Almohri, Yao & Kafura, 2014*) cannot guarantee the isolation of hijacked processes described under attacks. Therefore, it is generally advisable to monitor processes running on the device. If an unauthenticated process is launched, the process must be isolated to hinder damaging the device and analyze and detect the malicious application. This way, if a malicious application bypasses the detection barrier and downloads a malicious payload, it is caught when running an unauthenticated process to execute that payload.

(h) Triggering Malicious Code Assurance:

The process of ensuring the malicious code runs during the dynamic analysis sandboxing. TriggerScope (*Fratantonio et al., 2016*) statically tries to detect suspicious triggering; however, its limitation as static analysis makes it easy to be evaded by code obfuscation. Likewise, Groddroid (*Abraham et al., 2015*) developed a framework to launch the branches of each function to make sure that the malicious code starts. However, it fails to follow the components of background services, which misses the main activity. Groddroid is still an open issue among researchers and is known as code coverage. It is essential to address this issue by covering possible branches in the source code of the applications.

# CONCLUSIONS

Global evasion techniques make Android malware more advanced and sophisticated, which was our motivation for this study. We aim to highlight the most critical weaknesses of Android malware detection frameworks, mainly when malware uses different evasions techniques. Therefore, this study scrutinizes top Android malware detection frameworks against 18 evaluation test benches to evaluate the effectiveness of the evasions detection techniques in Android malware detection frameworks. Therefore, the study introduces a new evasion taxonomy that categorizes the evasion techniques into two main groups, polymorphism and metamorphism, where each group has branches; the polymorphism group includes package transformation, and the encryption metamorphism group contains code obfuscation, advanced transformation, and anti-emulation branches. The study also pointed out the lack of research in evaluating the malware detection against different evasion techniques; hence we scrutinized the frameworks based on every evasion technique and categorized the evaluations based on the malware detection methods. Our analysis results conclude a lack of research evaluating the current Android

malware detection framework robustness against state-of-the-art evasion techniques. We also concluded that static analysis based detection is easily evaded with simple obfuscation.

On the contrary, dynamic and hybrid analyses address advanced code transformation techniques and other advanced evasions. However, preliminary studies have evaluated their frameworks against evasion techniques. The missing framework evaluations are due to the lack of standard benchmark evasion datasets with updated standard malware datasets and the lack of comprehensive test benches tools to assess the efficiency of the existing and future frameworks. This study advises the research community to exert more effort into detecting anti-emulation evasion as indicated in the map of evasions and detection techniques. We also plan to create a standard evaluation framework to include all types of evasion techniques and consider the new generation of malware that combines multiple evasion techniques.

### Funding
This work was supported by Fundamental Research Grant Scheme under the Ministry of Education Malaysia (FRGS/1/2018/ICT03/UM/02/3). The funders had no role in study design, data collection and analysis, decision to publish, or preparation of the manuscript.

### Grant Disclosures
The following grant information was disclosed by the authors:
Fundamental Research Grant Scheme under the Ministry of Education Malaysia: FRGS/1/2018/ICT03/UM/02/3.

### Competing Interests
The authors declare that they have no competing interests.

### Author Contributions
- Wael F. Elsersy conceived and designed the experiments, performed the experiments, analyzed the data, prepared figures and/or tables, authored or reviewed drafts of the paper, and approved the final draft.
- Ali Feizollah conceived and designed the experiments, analyzed the data, prepared figures and/or tables, authored or reviewed drafts of the paper, and approved the final draft.
- Nor Badrul Anuar conceived and designed the experiments, authored or reviewed drafts of the paper, and approved the final draft.

### Data Availability
The cited papers (Endnote version 20.1) are available in the Supplemental File.

### Supplemental Information
Supplemental information for this article can be found online at http://dx.doi.org/10.7717/peerj-cs.907#supplemental-information.

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
