# Peer review of "The rise of obfuscated Android malware and impacts on detection methods"

_PeerJ Computer Science, doi:10.7717/peerj-cs.907_

## Round 0.1 · original submission · Major Revisions

Based on reviewers’ comments, you may resubmit the revised manuscript for further consideration. Please consider the reviewers’ comments carefully and submit a list of responses to the comments along with the revised manuscript.

Reviewer 1 ·

Basic reporting

There are fair amount of English language mistakes and ambiguous sentences. Some of them are highlighted and annotated in the pdf file. The intext citation style is inconsistent at places (highlighted in pdf)

Experimental design

The design of the study is satisfactory. However one of the major concern is that the study do not report the evaluation results of the surveyed papers. They should include table w.r.t evasion techniques and corresponding evasion results obtained in the literature. e.g. study A uses X evasion technique and successfully evaded n% of the sample under observation.
Moreover at multiple places in the article, headings are missing. (highlighted in the pdf).

Validity of the findings

Lessons learnt and the future directions are not presented clearly.

Additional comments

The paper needs major revision. Around 39 points are annotated in the pdf file which should be addressed clearly before publication.

Annotated reviews are not available for download in order to protect the identity of reviewers who chose to remain anonymous.

Reviewer 2 ·

Basic reporting

The English language should be improved to ensure that an international audience can clearly understand your text
• Incomplete sentence
o in Abstract at Line 20. For example, "Every day, thousands of new Android malware applications"
o Line 257, “Several untrusted or well-verified…”
• Too long sentences
o 1. Line 26 "The concern of encountering difficulties in malware reverse engineering motivates researchers to maintain the gap between securing the source code of benign Android applications by using evasion techniques and analyzing the obfuscated Android malware applications"
o 2. Line 38 "With many open-source libraries for Android....."
• Line 38 Since, add comma
• Line 46 e.g., contact lists, photos, videos, documents, AND account details
• Line 48 reference missing. "Android applications use Java as a developing language because ..."
o Line 54 reference Google Bouncer?
o Line 131, FeCO Reference?
• The word Obfuscation is use somewhere with capital O and somewhere with small o?
• The Rationale – The rationale correct it
• Line 52 ".;" correct it
• In line 23 "The malware authors adopt the obfuscations ".- Correct it
• Line 81 False Negative FN detection - False Negative (FN) detection, Line 83 use FN instead of abbreviating it again.
• Line 91 You and Yim (You & Yim 2010) – correct it
• Line 93 and polymorphic malware types
• Line 93 please explain the word these?
• Line 107, Correction required- “Our goal is to systematize these evasion techniques using a taxonomy methodology…” also explain the term ‘these’
• Line 152, we conclude the paper on conclusion section – correct it
• In line 188 the abbreviation WOS should be mentioned properly in Line 159
• Line 208, In the section, we first study the Android application – Correct it
• Line 209, we investigate the Android Operating System's (OS) weaknesses. – Correct it
• Repeating sentence check line 218 and 228. “Android apps require a virtual machine to run, called Dalvik or ART, depending on the OS version”
• Line 244, Therefore, The Android community, Correct it
• Formatting issues in Line 264 The Coarse Granularity of Android Permissions: label correction required, also explain this point to get clear understanding of android weakness.
• References missing in paragraph 270-274
• Line 344, 406, 410 and many others label missing, reference missing.
• Line 534 The classification techniques base the decisions on many criteria – correct it
• Figure ordering such as figure 7,8 mentioned after figure 9, 10

Experimental design

• Please explain in your introduction why you choose particularly Android platform for observing obfuscation techniques?
• Second and third contribution point (Line 110) should be to the point and give clear depiction of your work. Moreover, contribution point 1 and 2 overlap such as “This study examines different evasion techniques that hinder detecting malicious parts of applications and affect detection accuracy” and “Our goal is to systematize these evasion techniques using a taxonomy methodology, which clearly shows various evasion techniques and how they affect malware analysis and detection accuracy”
• Authors didn’t mention how many papers they investigated in number such as at Line 117 “our investigation focuses on studies written between 2011 and early 2021”
• Table 1 shows 11 comparison only, whereas authors have investigated studies from 2011-2021 so only 11 papers focused on evasion techniques? Or did authors mention only these papers, if so, why?
o In table 1 only two papers are from 2021, then 0 papers from 2020, 0 from 2019, 0 from 2018 indicates either there are no studies conducted in this era or authors didn’t surveyed properly.
• Line 131 “FeCO focused on features selections and machine learning models….” which models? And which type?
• Table 2, please fill publisher column for the remaining journals
• During identification and screening phase, authors didn’t use the term Machine learning, however, the title represent strongly as machine learning. And being a reviewer I didn’t found a word machine learning throughout introduction and related review section. Moreover, in line 660 author mention “However, deep insight into machine learning techniques is outside the scope of this study” then the title of the paper should be revised.
• Authors mentioned that they are the only one who scrutinize Android malware detection academic and commercial frameworks, so are you claiming that there are no studies that focused on academic framework?
• Line 284. “On the other hand, malware writers trap these features to gain remote access to install malicious applications”, Explanation required how?
• Line 307 Explain the word THEM and THESE in sentences “Android applications have powerful tools and techniques to secure and protect them from being reverse-engineered. Conversely, malware authors are using these tools and techniques to evade detection”
• Line 311, “As displayed in Figure 2, we categorize evasion techniques into two main types” There are other evasion techniques too, such as oligo-morphic etc. so did authors only considered polymorphic and metamorphic intentionally? If so, please explain why?
• Please extend the study by mentioning which papers focused on which evasion techniques through references. Such as Line 326 RPK, how many papers that you scrutinize uses RPK etc
• Most of the references used in the paper are old such as 2015-2016 and I barely found the updated references from 2019-2021, indicates that they survey is conducted only from papers 2014-2016.

Validity of the findings

No comment

Additional comments

no comments

---

## Round 0.2 · accepted · Accept

Congratulations, the reviewers are satisfied with the revised version of the manuscript and have recommended acceptance.

Reviewer 1 ·

Basic reporting

The basic reporting of the paper is up to the standards and all the references are concrete and relevant to the field of study.

Experimental design

Good

Validity of the findings

Fair

Additional comments

All concerns raised by my side have been addressed and i am happy to accept the paper for publication.

Reviewer 2 ·

Basic reporting

Required changes have been conducted. No comments

Experimental design

Required changes have been conducted . No comments

Validity of the findings

no comment

Additional comments

no comment